behaviour, cognition

reinforcement learning, advice, reputation, paranoia, trust, social learning

**Author for correspondence:**
Uri Hertz
e-mail: uhertz@cog.haifa.ac.il

# Trusting and learning from others: immediate and long-term effects of learning from observation and advice

Uri Hertz[1], Vaughan Bell[2] and Nichola Raihani[3]

[1]Department of Cognitive Sciences, University of Haifa, Haifa, Israel
[2]Department of Clinical, Education and Health Psychology, University College London, London, UK
[3]Department of Experimental Psychology, University College London, WC1H 0AP, London, UK

UH, 0000-0003-4852-3516

Social learning underpins our species's extraordinary success. Learning through observation has been investigated in several species, but learning from advice—where information is intentionally broadcast—is less understood. We used a pre-registered, online experiment ($n = 1492$) combined with computational modelling to examine learning through observation and advice. Participants were more likely to immediately follow advice than to copy an observed choice, but this was dependent upon trust in the adviser: highly paranoid participants were less likely to follow advice in the short term. Reinforcement learning modelling revealed two distinct patterns regarding the long-term effects of social information: some individuals relied fully on social information, whereas others reverted to trial-and-error learning. This variation may affect the prevalence and fidelity of socially transmitted information. Our results highlight the privileged status of advice relative to observation and how the assimilation of intentionally broadcast information is affected by trust in others.

## 1. Introduction

When learning about the world, individuals can either learn individually (through trial and error) or socially (from other individuals). Social information can be gleaned via two main routes: either by observing how others behave or by following explicit advice or recommendations. Although socially transmitted information is recognized as the basis for our species's extraordinary ecological success [1,2], we know relatively little about how individuals treat different kinds of socially transmitted information, how this varies with the need to evaluate demonstrators' abilities and trustworthiness, and the ways in which social information might affect behaviour over both the short and long term.

By copying others, individuals might shortcut the learning process [3,4] but the payoffs to copying are likely to depend on the demonstrator's abilities and expertise [3,5–8] as well as their preferences and values. Another way to learn from others is when someone intentionally broadcasts information, for example, in the form of recommendation or advice [9–12]. Advice is different from observational learning, as advisers broadcast information with the intention of being observed and influencing others' behaviour [13–15].

In addition, learning from advice may be privileged over observational learning because advice is perceived as providing more reliable information. Offering advice has potential implications for the adviser's perceived status [16–18]. Advisers who provide accurate information which benefits others might appear more knowledgeable, resulting in greater influence in the future and enhanced prestige [13,15,19]. On the other hand, making a mistake or giving the wrong advice can negatively impact an adviser's status and down-weight the probability that they will be copied by others [13,20–23]. The risk of losing prestige can

**Table 1.** Summary of model parameter estimation.

| | $\beta$ | $\alpha$ | Qboost$_{Advice}$ | Qboost$_{Observation}$ | $\beta$Boost$_{Advice}$ | $\beta$Boost$_{Observation}$ | log likelihood |
|---|---|---|---|---|---|---|---|
| min. | 0.1 | 0.05 | 0 | 0 | 0.1 | 0.1 | 2.11 |
| 1st quartile | 4.18 | 0.26 | 0.74 | 0.53 | 6.18 | 5.62 | 14.4 |
| median | 8.57 | 0.49 | 1 | 1 | 15.2 | 14.1 | 19.2 |
| mean | 15.2 | 0.52 | 0.85 | 0.80 | 25.0 | 23.5 | 18.9 |
| 3rd quartile | 19.5 | 0.80 | 1 | 1 | 50 | 50 | 24.0 |
| max. | 50 | 0.95 | 1 | 1 | 50 | 50 | 31.1 |

affect advice-giving strategies: for example, advisers might offer advice only when they believe that their information is accurate [14,17,24]. The fact that advisers are accountable for the information they provide can make advice more reliable than information that is gleaned by eavesdropping [25]—and receivers may therefore be more willing to modify their behaviour in response to advice than in response to observed social information.

However, the interests of advisers and receivers may not always align [26]. When interests are mismatched, advisers may try to deceive the receiver such that the receiver takes an action that is to the adviser's benefit [24,27,28]. The risk of being deceived may be most prevalent in environments with high uncertainty, or where there is a big gap in knowledge between the adviser and advisee, which make deception harder to detect [20,26,27]. Suspicion about the intentions of others, and their benevolence, may therefore impede learning from advice [10,29,30]. An exaggerated tendency to believe that others have malign intentions—and associated suspicion and mistrust—is the basis of paranoia [31,32], which is not solely a clinical category but also varies across a full spectrum of severity in the general population [31,33,34]. Given the increased tendency to believe that others have harmful intentions, even when true intentions are ambiguous [35,36], we might expect higher levels of paranoia to be associated with lower levels of trust in others and, consequently, a reduced tendency to follow advice. Importantly, no such patterns should be observed in the realm of eavesdropping, since a demonstrator's intentions are not a relevant concern in this scenario [37,38].

The effect of social information on a receiver's behaviour might also vary over the short and long term. For example, a receiver might initially follow social information but revert to using personal experience over the longer term [12,17]. Alternatively, social information could have a longer term effect on receiver decisions [39]. When the environment is stable, and social information is reliable, long-term adherence to social information, may be beneficial to the individual. This is captured by producer-scrounger dynamics, where a producer explores the environment and a scrounger exploits the information uncovered by the producer [40,41]. The longevity of social information use, and its dependence on the source of social information, can affect social transmission and the perseverance of social information in groups [42], as long-lasting effects are more likely to be transmitted to others.

To test these hypotheses, we designed a two-armed bandit task [43,44] framed as a fishing task (figure 1) in which participants had to identify the best lake in each trial and in which they were exposed to three social information conditions:

control (no social information provided), observation (observe the choice of an expert player) and advice (receive advice from an expert player). Participants chose between two lakes over 15 trials and obtained feedback about the reward obtained after each decision. In all conditions, participants made choices without social information over the first four trials, to allow them to establish some expectations of the lakes. This was useful for q-learning modelling (described below), which explored the effect of previous experience on advice taking. In the observation and advice conditions, social information was presented once, after four trials and always suggested the higher paying option. This allowed us to examine the effect of social information immediately after it was presented (in trial 5) and also over the long term (trials 6–15). We were especially interested in the difference between following advice and copying observed choices, as well as the interaction of paranoia with these effects. Social information was always accurate (indicating the lake with the greater number of expected fish) to avoid the introduction of irrelevant differences between social information conditions, but was not always in line with the experience-based expectations of the participants, and did not always lead to immediate high rewards. Our main hypotheses (detailed below in the methods section) and analyses were pre-registered with additional exploratory analyses reported in the text.

## 2. Methods

### (a) Participants

All data were collected in July 2020. We recruited 1498 participants (592F, 895M, 11 classified themselves as non-binary, other or did not report gender) from Prolific Academic. Participants' mean age was 29.2 (s.d. = 10.4). This sample size was set to allow us to sample effectively across the full distribution of paranoia levels, based on previous studies [35,45] and was pre-registered. All participants provided informed consent and received monetary compensation at a fixed rate of £1.50 GBP for participation and could gain up to £1.80 as a performance-based bonus (average bonus = £0.91). On average, the task took 17 min to complete and participant earnings were equivalent to £8.22 per hour in the task. All participants received a bonus (minimum £0.80). No participant was excluded from analysis, but single trials that took more than 20 s to complete were removed, resulting in the removal of 0.3% of trials (199/67 140) across all experimental conditions and learning trials. This procedure was not pre-registered, as we did not anticipate long response times based on the pilot study. We decided to exclude these long trials to remove trials which we thought did not reflect

(*a*)

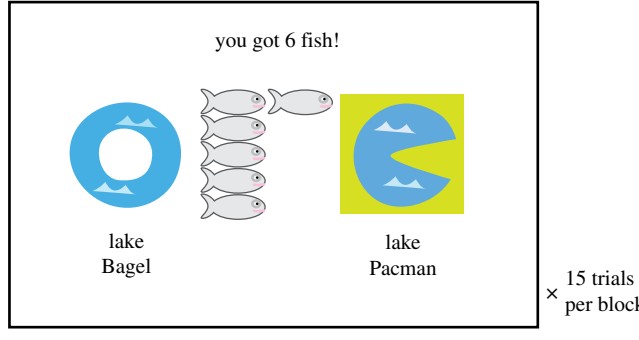

(*b*)

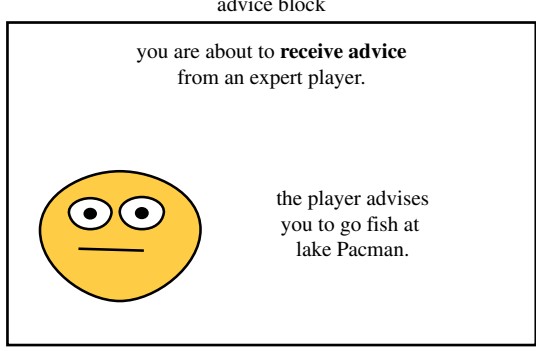

**Figure 1.** Experimental design. (*a*) Participants completed three experimental blocks of the fishing task. On each trial, they had to choose between two lakes to fish from. They received immediate feedback about the number of fishes they caught. (*b*) Participants received social information after the 4th trial in two experimental blocks. (Online version in colour.)

genuine task performance. We note that an analysis without trial exclusion did not lead to any meaningful changes in the results.

## (b) Experimental design

In the fishing task (figure 1), participants ($n = 1492$) chose to fish at one of two lakes and subsequently received feedback about the number of fishes they caught. Participants made this choice 15 times for each pair of lakes. The number of fishes caught was randomly drawn from a normal distribution, with one lake having higher average yield than the other ($M_{Good} = 5.5$, $M_{Bad} = 4$), and both lakes having the same variance ($\sigma = 1.7$). As these distributions overlapped, it was possible to receive high rewards from the bad lake and vice versa in some trials. At the end of the task, participants were paid a bonus that was determined by the total number of fishes they caught (20 pence per 50 fishes caught; average response time and reward for each block are in electronic supplementary material S10). The experiment was programmed in JavaScript, and the code is available (see 'Data accessibility' below).

Each participant completed three blocks: control, observation and advice (order counter-balanced), with each block differing in the social information provided to participants after the 4th trial. Social information always recommended the good (high expected reward) lake. However, this was not immediately apparent to the participants, as their initial experience in the first four trials could contradict the social information (see analysis of initial experience in the electronic supplementary material, figure S1). Participants subsequently completed a paranoia questionnaire (the revised Green *et al.* Paranoid Thoughts Scale—R-GPTS [46], which includes measures of persecution-paranoia and reference-paranoia levels) and a short fluid intelligence test, the short Hagen Matrices Test (HMT-S) [47]. Following the guidance in Freeman *et al.* [46], we used the persecution subscale of the R-GPTS as a measure of paranoid thinking.

## (c) Pre-registered hypotheses and analyses

Hypotheses and analyses conducted in this study were pre-registered (https://aspredicted.org/blind.php?x=q49c3q). Analyses that were not pre-registered are clearly stated. Some pre-registered analyses are included in the electronic supplementary material only. We pre-registered five hypotheses concerning the effects of social information on learning:

H1. We expected that participants would be more likely to follow advice than observed social information. We measured this by recording choices in trial 5 (the trial that occurred immediately following information presentation). The decision to choose the good lake (1/0) in trial 5 was specified as the response variable in a mixed-effect logistic regression. Our pre-registered dependent variables included the experimental condition (within-subjects) (advice/observation/control), paranoia [46] and fluid intelligence score [47]. We also included interactions between paranoia, and experimental conditions, and the interaction between intelligence and experimental conditions.

H2. We expected that overall performance (number of times participant chose the best lake) would be increased in the advice (relative to the observation and control) condition.

H3. We expected that receiving advice would result in participants converging more quickly to the good option (H3). This was determined by measuring the number of trials that elapsed before the participant chose the good lake in three consecutive trials.

H4. We expected that choice stochasticity would increase with paranoia. We used the U-value measure of choice stochasticity [48] as a dependent variable in a mixed-effect linear regression with explanatory terms experimental condition and paranoia.

H5. We expected to detect the effects of social information in a post-learning preference test, where lakes indicated by advice would be more likely to be identified in a post hoc preference test than those indicated by observation.

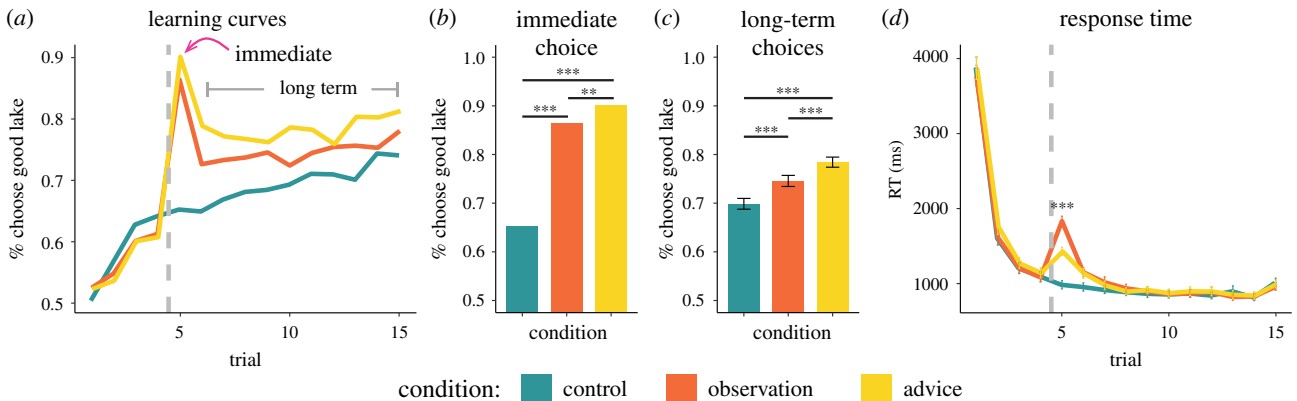

**Figure 2.** Immediate and long-term effects of social information on choice. (*a*) The percentage of participants choosing the good lake increased over trials. Dashed lines represent the trial in which participants received social information (or no social information in the control condition). Immediate and long-term improvements in performance resulted after social information presentation relative to the control condition. Participants were more likely to choose the good lake after receiving advice compared to eavesdropping both immediately (*b*) and in the long term (*c*). (*d*) Participants were slower to choose a lake after receiving social information and significantly slower to choose after observing another player than after receiving advice. Bar heights represent mean values; error bars in (*c*) and (*d*) represent 95% confidence intervals. \*\*\**p* < 0.0005, \*\**p* < 0.005. (Online version in colour.)

In addition to these main hypotheses, we suggested that increasing paranoia would be associated with a decreased tendency to follow advice (but would not be associated with a tendency to copy observed behaviour) and that performance in the advice condition would therefore be impaired in paranoia (H1a, H2a, H3a, H5).

## (d) Analysis

We used mixed-effects linear and logistic regressions with group-level coefficients (also known as fixed effects) to model the population-level effects and individual-level coefficients (also known as random effects) to capture individual average responses [49]. Our models included within-subjects effects related to the task, such as the experimental condition and block order, and effects that varied between subjects, such as paranoia levels and intelligence scores. To estimate the regression effects, we used Type III Wald chi-square tests [50]. We estimated the population-level marginalized means for post hoc evaluation of the effects and for post hoc comparisons [51]. Computational learning-model fitting was carried independently for each participant using in-house code and L-BFGS-B constrained quasi-Newton optimization method, implemented in R optim function [52]. Analysis software is detailed in the electronic supplementary material, all code and scripts are available online.

## 3. Results

### (a) Immediate effect of social information

Participants were more likely to choose the good lake following advice compared to when they simply observed another player's choice (Type III Wald $\chi^2$ test: $\chi^2 = 45.9$, $p < 0.001$; figure 2*a,b*; electronic supplementary material, table S1). As predicted, paranoia was negatively associated with the likelihood of following advice, but was not associated with the likelihood of copying observed choices ($\chi^2 = 6.18$, $p = 0.045$; figure 3*a*).

In an additional unregistered exploratory analysis, we examined the effect of social information on response time in the trial following the social information (figure 2*d*; electronic supplementary material, table S6). Speed of response was affected by the condition ($\chi^2 = 47.06$, $p < 0.001$), with participants responding faster when following advice compared to when copying an observed choice ($t_{2992} = -10.2$, $p < 0.001$).

This suggests that participants deliberated less when following social information framed as advice compared to when using social information framed as observation.

### (b) Long-term effect of social information

To examine the long-term effects of social information on performance, we calculated the average number of trials in which participants chose the good lake over trials 6–15. As expected, participants chose the good lake more often in the advice condition than in either the observation or the control conditions ($\chi^2 = 28.5$, $p < 0.001$; figure 2*c*; electronic supplementary material, table S2). Nevertheless, our prediction that long-term performance would be impaired among more paranoid individuals in the advice condition was not supported (figure 3*b*). Average performance over the long term was highest in the advice condition (0.78 [0.77, 0.79]), lower for the observation condition (0.75 [0.73, 0.76]) and lowest in the control condition (0.70 [0.69, 0.71]) (figure 2*c*). The long-term effect of receiving advice on performance was significantly higher than the long-term effect of observing another person's choice ($t_{2992} = 4.8$, $p < 0.001$).

### (c) Modelling effects of social information on learning and choice

Next, we analysed the participants' learning patterns, looking at the time to reach convergence in choices and choice stochasticity (H3, H4). As expected, participants reached convergence (defined as choosing the better lake in three consecutive trials) earlier in the advice condition compared with observation and control conditions ($\chi^2 = 8.63$, $p = 0.013$; electronic supplementary material, table S3). Participants with high levels of paranoia were marginally slower to reach convergence, regardless of experimental condition ($\chi^2 = 3.82$, $p = 0.05$). Choice stochasticity (u-value) [48] was affected by the experimental condition ($\chi^2 = 30.0$, $p < 0.001$), with participants showing lowest choice stochasticity in the advice condition compared with observation and control conditions, and was higher among more paranoid individuals ($\chi^2 = 8.60$, $p = 0.003$; electronic supplementary material, table S4). These models further support the inference that advice improves performance in the learning task.

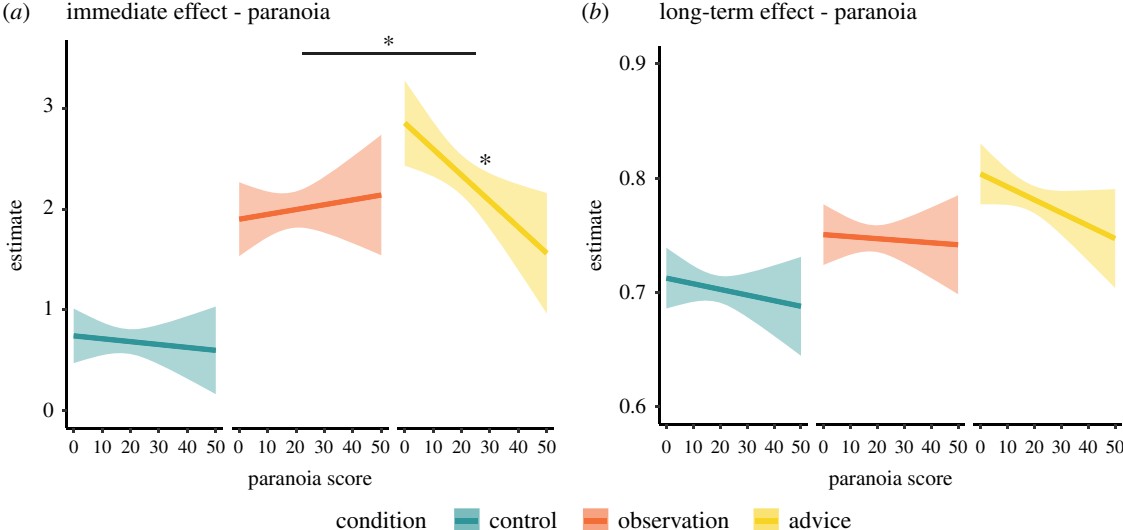

**Figure 3.** The effects of self-reported paranoia on assimilating social information. (*a*) Paranoia was negatively associated with the following advice, but did not affect the tendency to copy observed choices or performance in the control condition. (*b*) Paranoia was not associated with long-term effects on performance. Lines are the estimated marginal trends from the regressions, and shade areas represent 95% confidence of the estimation. *$p < 0.05$. (Online version in colour.)

To gain a more mechanistic evaluation of the immediate and long-term effects of social information on learning, and to expand on the analyses of convergence rate and choice stochasticity, we used a computational reinforcement q-learning model (following [7,8,39,53]). This approach was pre-registered as an exploratory analysis. In this model, the value of each lake was learned and updated on a trial-by-trial basis by comparing the expected reward and the actual reward gained from the lake (the prediction error). The expected value update rate was determined by a learning rate parameter $\alpha$:

$$Q_{\text{LakeA}}(t + 1) = Q_{\text{LakeA}}(t) + \alpha \cdot (\text{Fish}(t) - Q_{\text{LakeA}}(t)). \quad (3.1)$$

The choice of lake on each trial was determined by a softmax rule, which considers the relative expected value of the lakes (their Q-values). A precision parameter, $\beta$, indexes how deterministic decisions are: higher $\beta$-values imply more deterministic choice, which can be understood as choosing even marginally better options with higher probability.

$$p(\text{Choose Lake } A) = \frac{\exp(\beta \cdot Q_{\text{Lake}A})}{\exp(\beta \cdot Q_{\text{Lake}A}) + \exp(\beta \cdot Q_{\text{Lake}B})}. \quad (3.2)$$

To account for the immediate learning effect, the model included two free parameters, $\{Q\text{boost}_{\text{Advice}}, Q\text{boost}_{\text{Observation}}\}$, that determined the value of the option indicated by social information. When these parameters are set to 1, the value of the lake indicated by the social information reaches maximum level in the trial after advice. Although this immediate boost value may exert a longer-term effect, it may also decay over subsequent trials. To account for the potential long-term effects of receiving social information, the model included an additional two free parameters, $\{\beta\text{boost}_{\text{Advice}}, \beta\text{boost}_{\text{Observation}}\}$, which were used to set the value of the precision parameter $\beta$ after the social information is given. When these parameters are high, participants' choices follow a 'greedy' decision rule and they become exploiters, as they always choose the lake with a higher $q$-value even when the difference between lakes is small [54]. Low $\beta$boost values indicate a more exploratory behaviour, where there is a substantial possibility of choosing the lake with a lower Q-value. Both changes to

precision and value took place only once in our model, immediately after receiving the social information, in line with our experimental design.

We simulated this model to examine whether it could reproduce the immediate and long-term effects of social information on decisions (figure 1*a*). First, we simulated the model using different values for the $Q$boost parameters, showing that an increase in the Q-value of the good lake immediately after advice produces an immediate but short-lived effect (figure 4*a*). We then simulated the model using different values for the $\beta$boost parameters to examine the effect of social information on decisions over the longer term. These simulations demonstrated that increasing the $\beta$-values after social information produced a long-term effect of convergence to the good option (figure 4*b*). However, a similar long-term effect as simulated by varying the values of $\beta$boost parameters could also come about if the population comprised a mixture of participants with different $\beta$boost values.

Specifically, we explored scenarios where our participant pool contained different ratios of participants with high (exploiters) and low (explorers) $\beta$boost values (figure 4*c*). As the ratio of individuals with high $\beta$boost values increased, the long-term effect of social information also increases. For example, specifying a population where 40% of participants have high $\beta$boost values could reproduce the group effect we observed in the participants' behaviour in the advice condition (figure 2*a*). Although both approaches—assuming a unitary $\beta$boost value for the entire participant pool or assuming a bimodal $\beta$boost value across the participant pool—could account for the observed group effects, they imply stark differences in the long-term effects of social information on subsequent choices.

To test which simulation best described participants' behaviour, we fitted the computational model to each individual and estimated the free parameters for each participant:

$$\{\beta, \alpha, Q\text{boost}_{\text{Advice}}, Q\text{boost}_{\text{Observation}}, \beta\text{Boost}_{\text{Advice}}, \beta\text{Boost}_{\text{Observation}}\}.$$

In line with our simulations, the immediate value boost parameters were very close to one for most of our

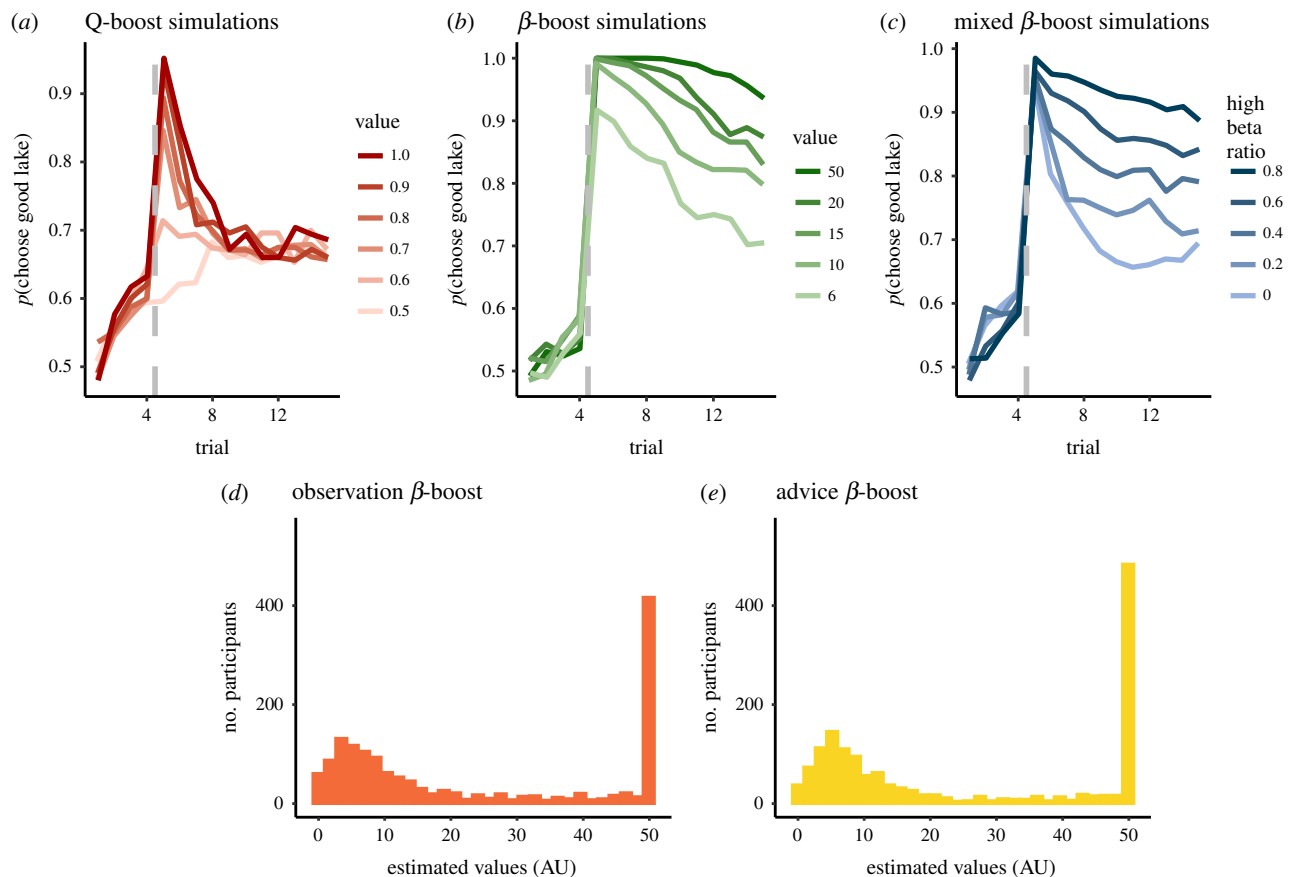

**Figure 4.** Model simulations for immediate and long-term effects of social information. (*a*) We simulated our model with a different value for *Q*boost parameters, which increases an option's Q-value immediately after social information to generate the immediate effect of social information. (*b*) We simulated the model with different values of *β*boost parameter to obtain the long-term effect of social information. (*c*) We simulated the model with different ratios of two populations, high and low *β*boost and low beta boost, and obtained long-term effects. After fitting the model, we found that the distributions of the precision parameters *β*boost for observation (*d*) and advice (*e*) followed a bimodal pattern, with around 40% of our participants having the maximum *β*boost value following social information. (Online version in colour.)

participants and were higher on average for advice than for observation (table 1), implying that for most participants the social information increased the value of the suggested lake to the maximum level. Examination of the distribution of estimated values of *β* and *β*boost parameters revealed two distinct peaks (figure 4*d*–*f*), indicating that our participant pool comprised two types, with respect to the long-term effect of social information on choices. We classified participants as high when their estimated *β*boost parameter was close to ceiling (above 40, when the ceiling was 50), and low otherwise (figure 5*a*). In line with the mixed population simulations, we found that 39% of our participants were estimated to be high *β*boost in the advice condition compared with 34% in the observation condition. These populations overlapped, such that 18% of participants displayed high *β*boost both in the advice and observation condition. We found that participants in all sub-groups tended to display an immediate effect of advice. However, the display of consistent long-term choice fidelity to the good lake after social information was associated with high *β*boost values (figure 5*a*). This pattern was distinct from the more variable pattern associated with lower values, which may include more elaborate learning trajectories which are dependent on the specific outcomes experienced by participants in this group. Participants with high *β*boost, who exploited social information throughout the learning block,

made their choices significantly faster ($\chi^2 = 10.39$ $p = 0.0012$, figure 5*b*; electronic supplementary material, table S8) and gained significantly more fishes ($\chi^2 = 224.89$ $p < 0.0001$, figure 5*c*; electronic supplementary material, table S9) than participants with low*β*boost, and as a result received higher bonus payments while spending less time performing the task.

Finally, we examined factors affecting variation in *β*boost values in the advice and observation conditions (electronic supplementary material, table S7). *β*boost value was affected by block order ($\chi^2 = 51.3$, $p < 0.001$), and its interaction with block type ($\chi^2 = 10.31$, $p = 0.001$). Specifically, participants' likelihood of exploiting observed social information over trials 6–15 (high *β*boost) was higher when the observation condition occurred later in the experiment. This order effect was much attenuated for the advice condition (figure 5*d*), indicating that the likelihood of exploiting advice over the long term was not affected by the order in which the participant played the advice condition. These two results suggest that the long-term effect of advice was not dependent upon experience with the task, whereas the tendency to assimilate observed information over the long term might have been affected by familiarity with the task. More paranoid individuals had lower *β*boost values ($\chi^2 = 5.46$, $p = 0.02$), which were not specifically linked to advice or observation (figure 5*e*). This supports the inference that paranoia is associated with

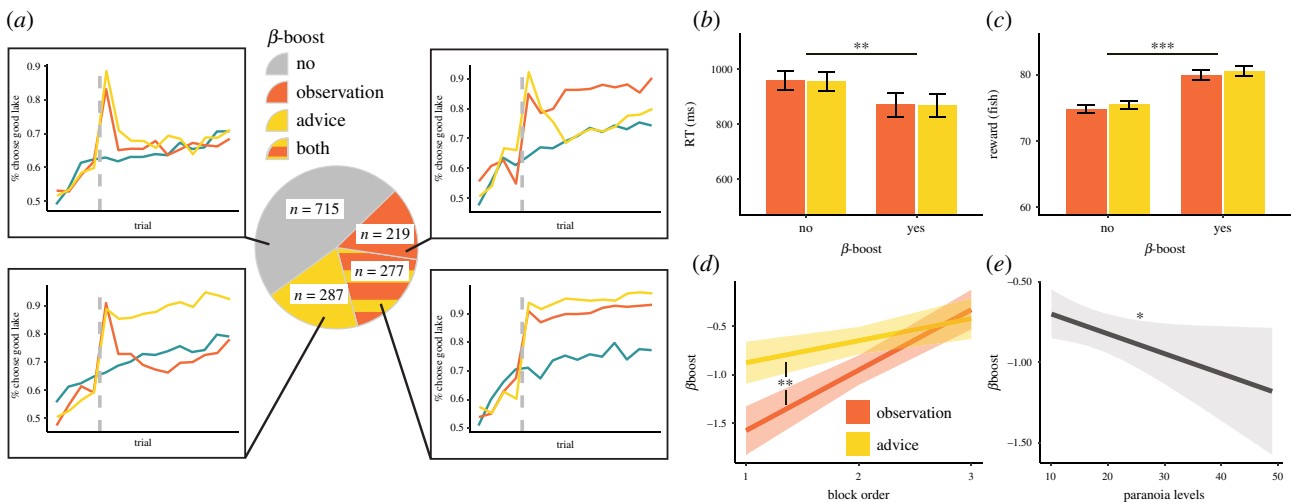

**Figure 5.** (*a*) Four distinct patterns of long-term effect of social information. We used the estimated $\beta$boost parameters to classify participants into one of four types—no $\beta$boost, observation $\beta$boost, advice $\beta$boost and both $\beta$boost. We averaged the choice pattern in all four groups and found that high $\beta$boost was associated with almost complete convergence to the good option after social information, while low $\beta$boost was associated with return to baseline learning pattern, similar to the control condition. This is in line with a bimodal account of long-term effect of social information. Participants with high $\beta$boost spend less time for each choice (*b*) and received higher rewards (*c*). (*d*) The likelihood of having high $\beta$boost after an observation was related to the block number within the experiment. There was no such association for the advice $\beta$boost. (*e*) There was an overall negative relation between high $\beta$boost and paranoia levels. Lines are the estimated marginal trends from the regressions, and shade areas represent 95% confidence of the estimation. **$p < 0.005$, *$p < 0.05$. (Online version in colour.)

increased choice stochasticity and with taking longer to converge to the good option.

## 4. Discussion

We show that people were more willing to follow advice than to copy an observed choice and that both immediate and long-term performance was higher after participants received advice, compared to when they simply observed the choice of another player. As predicted, advice-following was dependent upon trust in others, being reduced in paranoia. This latter result may be interpreted in line with concerns about others' intentions, which are likely to be more salient when following advice compared to when copying observed decisions. Supporting this interpretation, we detected no association between paranoia and the tendency to copy observed decisions. In addition, people followed advice more quickly than they imitated an observed choice. By fitting a computational model to our data, we were able to more intensively interrogate patterns of learning, both immediately after receiving social information and over the longer term. Our sample was found to be bimodal with respect to the long-term effects of social information on behaviour: some participants continued to choose a previously advised or observed decision over all subsequent trials, while others used social information in the short term, but reverted to baseline thereafter. This bimodal distribution has two important features. First, it indicates that participants either fully exploit social information or stick with a trial-and-error strategy, and do not show an intermediate effect of slightly increased tendency to exploit social information. Second, it indicates that responses to social information are not unified across the population (or don't follow a unimodal distribution). Taken together, these features may have implications for modelling and understanding social information transmission dynamics.

Participants were more likely to immediately follow advice than to immediately copy another player's choice, suggesting

that advice is evaluated as being a more reliable source of information than observed actions, in line with previous studies [42,55–57]. One reason for this may be because there is a reputational risk associated with intentionally advising others how to behave. Previous studies investigating learning from observation [3] and even from demonstration [58] indicate that evaluating the abilities of an observed agent is cognitively demanding [21]. It is possible that advisers' desire to appear competent serves as an epistemological mechanism, which may either partially or completely obviate the need for receivers to evaluate an adviser's abilities or expertise [25]. Our finding that participants were faster to follow advice than to copy an observed decision suggests that participants deliberated less when following advice, suggesting that copying observed choices may involve a cognitively demanding evaluative process. The relationship between advice-following and paranoia also supports the role of trust in an adviser's intentions in social learning from advice. In previous studies, participants with high levels of paranoia did not differ from the general population in revising their choices after observing others' choices [37,38]. Our findings constitute a conceptual replication of this effect (in the observation block), but suggest that the effects of paranoia on social learning might depend on the means by which information is transmitted.

Our study differs from previous studies of advice giving in the way that we assume that individuals learn from social information over the longer term. In previous studies, the impact of information was modelled by a one-time increase in the value of the advised option (equivalent to our Qboost parameter) and in biased learning from the following information [11,39]. For the latter, the subjective value of rewards was higher when the rewards stemmed from a previously advised choice. These studies, which used a different task (four-armed bandit) and a smaller sample size, reported that advice had a unimodal long-term effect on choice, increasing the likelihood of choosing the advised option over time [11,39]. Although our model used a similar one-time value increase to the advised option, it differed in the mechanism

driving the longer-term effect: rather than a biased subjective reward, we assume that social information led individuals to increase precision in choices, an effect which controls the level of exploration and exploitation in the task [59].

Our observation that long-term social learning strategies are bimodally distributed is in line with earlier studies of social learning in humans and other animals. Other studies of social learning also indicate that social learning strategies vary across context and populations [5,60–62]. Individuals vary in the extent to which they learn from others, with some sub-groups of participants consistently copying from others or conforming with group's behaviour and others being more likely to learn through trial and error [60,63]. In animal behaviour, a similar bimodal distribution was observed in the context of the producer–scrounger distinction, which was modelled extensively using evolutionary game theory [41,64–66]. Under this logic, some animals exhibit a producer strategy, for example, initiating exploration for new food resources, while other animals exhibit a scrounger strategy, by exploiting the information revealed by the actions of producers [67]. To our knowledge, no studies of producer-scrounger dynamics in non-human species have explored whether information gleaned through advice (e.g. alarm-calling, teaching or food-related calls [68–70]) is treated differently to information that is transmitted via observation, nor whether the effects of such learning might vary over the short and long-term. Our findings that exploitation was more prevalent after receiving advice, and that this effect was less dependent on previous experience in the task than the effect of observation, suggest that advice is privileged in its long-term effects. We believe this would be a fruitful avenue for further exploration, as one might predict that intentionally broadcast information might generally be more likely to be faithfully transmitted, copied and assimilated, except for in cases where the adviser has an incentive to deceive [71].

Our distinction between two forms of social information (advice and observation) can also be cast in more general terms to be linked with descriptive and experiential information [72,73]. The expected value of an option can be described explicitly, or learned through experience. Recent work has shown that participants rate options differently, according to whether they obtain the information from description or through personal experience [73]. Specifically, the subjective likelihood of a rare event is exaggerated when rare events are explicitly described, but underrated when their prevalence is learned from experience [74]. The framework of the description–experience gap may be useful when evaluating the distinction between advice and observational information. Advice and explicit description stem from a knowledgeable person that intentionally shares the information, and therefore are similar in their epistemic source, while experience and observation credibility stem from one's own evaluative process. Importantly, many experiments in decision-making highlight the role of the experimenter as the one providing the explicit description of options (and the one in charge of programming the computer-based task), and the way trusting the intentions of the experimenter affect behaviour [75]. In the light of these differences, the advice–observation gap may have considerable methodological implications for interpreting findings about social learning.

There are a number of limitations to be borne in mind when interpreting these data. First, the data all came from UK-based participants recruited via Prolific Academic, so it is not clear how these findings generalize to other groups and contexts. One of the limitations of experimental designs in general is that they are an engineered situation that may not fully reflect the complexity of the phenomenon as it occurs more broadly. In this respect, our study has much in common with other experimental economic games, where experimentally controlled social scenarios are used to glean insights about social preferences and behaviour. Here, the goal of experimental economics should be seen as understanding the directional effect of experimental factors on social preferences and behaviour, under the assumption that (*ceteris paribus*) the same direction of effects will be observed in the real world [76]. We also note that in our study, social information was always on-average informative, which does not reflect the potential range of accuracy of advice in broader social situations. Importantly though, the fact that social information was only statistically informative and did not always lead to immediate rewards meant that this informative nature of advice was probably not apparent to all participants due to the stochastic nature of the learning task (in other words, negative outcomes could occur after following advice). This decision was taken to allow for analytical tractability. Future work will explore how information reliability affects how people use (and continue to use) information received via advice versus observation. A final limitation is that we did not ask participants whether they perceived information gleaned via advice versus observation to be differentially reliable, or whether they felt obliged to follow the social information. Although participants were more likely to follow advice, some or all of this tendency could have been explained by a normative expectation that advice should be followed. Although our finding that advice-following was reduced in paranoia mitigates against this explanation, we cannot entirely rule it out.

To conclude, we examined how framing social information as observation or advice affected learning. Our results showed that advice has a greater immediate and long-term effect on learning. Importantly, people with high levels of paranoia were less likely to follow advice, and we suggest that this is because advice is imbued with intent, meaning that receivers must trust advisers before the following advice. Our findings highlight that actively provided social information (advice) is treated differently to passively transmitted social information (eavesdropping). This distinction may lead to better characterization of information sharing in humans and other animals.

**Ethics.** This study was approved by the UCL ethics committee (project no. 3720–002). All participants provided informed consent and received monetary compensation.

**Data accessibility.** Data, experiment scripts and analysis scripts are available here: https://osf.io/kxwtn/?view_only=535795aae26447c0b28e6ee1426477fc.

**Authors' contributions.** U.H.: conceptualization, formal analysis, methodology, visualization, writing-original draft, writing-review and editing; V.B.: conceptualization, writing-original draft, writing-review and editing; N.R.: conceptualization, funding acquisition, resources, validation, writing-original draft, writing-review and editing. All authors gave final approval for publication and agreed to be held accountable for the work performed therein.

**Competing interests.** We declare we have no competing interests.

**Funding.** U.H. was supported by the Israel Science Foundation (1532/20) and by the National Institute of Psychobiology in Israel (211-19-20). N.R. was supported by a Royal Society University Research Fellowship (UF160412) and The Leverhulme Trust.

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
