## [Peer Review File · Proceedings of the Royal Society B: Biological Sciences]

Review History

RSPB-2021-1414.R0 (Original submission)

Review form: Reviewer 1

Recommendation

Accept with minor revision (please list in comments)

Scientific importance: Is the manuscript an original and important contribution to its field?

Good

General interest: Is the paper of sufficient general interest?

Good

Quality of the paper: Is the overall quality of the paper suitable?

Excellent

Is the length of the paper justified?

Yes

Should the paper be seen by a specialist statistical reviewer?

No

Do you have any concerns about statistical analyses in this paper? If so, please specify them explicitly in your report.

Yes

It is a condition of publication that authors make their supporting data, code and materials available - either as supplementary material or hosted in an external repository. Please rate, if applicable, the supporting data on the following criteria.

Is it accessible?

Yes

Is it clear?

Yes

Is it adequate?

Yes

Do you have any ethical concerns with this paper?

Yes

Comments to the Author

I really enjoyed this manuscript. I commend the authors' dedication to open and transparent research practices; pre-registering their hypotheses and making all data and analyses openly available. All papers are easier to review when this is the case.

I think the study is very interesting, combining paranoia with social learning tendencies is definitely a gap in the literature and an important avenue for exploring individual differences in cognition and how they impact social learning. Also the use of "advice" versus "observation" is definitely underexplored and worth more investigation.

I think the task is also really fun and engaging for participants. And beautiful, clear figures. I like the combination of experiment and model also, although I have to say here that I am awful at understanding models, this is a big gap in my knowledge. So I am not going to comment on the modelling, please find someone else to do that. There are plenty of other social learning modellers out there who would be much better at examining this than me (E.g. Elena Miu, Tom Morgan, Bret Beheim, Paul Smaldino, Anne Kandler...).

Mainly I just have a few conceptual clarifications and extra details to strengthen the manuscript.

Conceptual clarifications:

Prestige v reputation:

I like your discussion of reputation as a mechanism for replacing a more cognitively demanding evaluation of success. This fits very nicely with recent work on prestige bias, and I wonder whether you see reputation as a kind of prestige, and if so, it is worth mentioning that I think. I think it would be worth engaging with the more recent prestige literature than the cited Atkisson 2012 paper. E.g.

Brand, C. O., Mesoudi, A., & Morgan, T. (2021). Trusting the experts: the domain-specificity of prestige-biased social learning.

Brand, C. O., Heap, S., Morgan, T. J. H., & Mesoudi, A. (2020). The emergence and adaptive use of prestige in an online social learning task. *Scientific reports*, 10(1), 1-11.

Jiménez, Á. V., & Mesoudi, A. (2019). Prestige-biased social learning: current evidence and outstanding questions. *Palgrave Communications*, 5(1), 1-12.

Cheng, J. T., Tracy, J. L., Foulsham, T., Kingstone, A., & Henrich, J. (2013). Two ways to the top: evidence that dominance and prestige are distinct yet viable avenues to social rank and influence. *Journal of personality and social psychology*, 104(1), 103.

For example, in light of the Scientific Reports paper above, it can't be the case that reputation is necessary to "obviate the need of receivers to evaluate the demonstrator's abilities or expertise" (line 378), as this is achieved by prestige-bias; instead of having to evaluate expertise/success yourself, you can use prestige cues that have emerged via the social learning dynamics themselves. Unless you would view someone who is "most copied/paid attention" as a type of reputation in the way you discuss it? Maybe this is in fact what you mean, either way I think it would help to more explicitly mention how you think reputation does/doesn't differ from prestige, and reference the interplay between prestige and advice-giving.

Advice:

Similarly, it's not made entirely explicit or clear in the introduction / hypotheses why you think advice will be used more than observation, you have to sort of read between the lines.

Related, I like the penultimate discussion paragraph about your own evaluation of information versus another's evaluation. I think it's worth mentioning though (maybe in this paragraph or elsewhere) that if someone is giving advice, it is presumably implied/inferred that this advice is an amalgamation of (usually many) different experiences. It would be odd for someone to give advice based on just one dip in the lake, whereas in the observation condition you are just viewing one dip in the lake. It's possible that participants assumed that the advice-giver had finished their session of lake dipping, and based on all their dips, they are advising this lake. I know participants were never explicitly told that the advice-giver had taken more than one dip, but what were they told? And what might they have inferred? It would be nice to ask them post-experiment, I know it's too late for that now so maybe just mention that one possible explanation is that once information is given in the form of advice, it may be assumed that it is made up of more experiences than just one.

Trust :

Similarly in the discussion you mention "trust" a lot in the abstract and again in the first discussion paragraph, but you didn't actually measure trust in the demonstrators, right? Or are you assuming trust to be wrapped-up in the paranoia spectrum? Isn't it possible that two people on the same place on the paranoia spectrum might have different levels of trust in one particular model for different reasons? E.g. two people might be high in paranoia, but for some reason they have different levels of trust in Adviser A compared to Adviser B? Please elaborate on this.

Eavesdropping and bad intentions:

I follow the logic of why a paranoid person may trust what someone says *to them* less than what they see that person do "secretly" or through "eavesdropping," but I'm not entirely convinced. Particularly line 70-71 in the intro, and mentioned again in the discussion, I'm not entirely convinced a paranoid person would trust someone if they were eavesdropping more than when they were being told explicitly. Surely if I am paranoid and I believe people have bad intentions, then a person is just as likely to be lying to someone else (while I secretly listen to them) as they are likely to be lying to me when they directly address me? But I am no specialist in paranoia so please correct me if I've misunderstood (or elaborate on this slightly in the introduction for people outside the area).

Reliability of social info:

Line 88 - Why was social information always reliable? You mention to avoid differences between the two social learning conditions, but surely they could both vary in reliability to the same degree? It wasn't mentioned in the pre-reg either so it just seems a bit odd to make social learning unrealistically reliable. But maybe it was simpler, and you can't change it now, so perhaps just mention that this is of course unrealistic but was for simplicity, and mention this as a limitation in the discussion. E.g. If you knew someone's advice was always 100% accurate - of course you'd take it! I know the interesting bit then is then why advice is preferred over observation, but it's possible that if they both fluctuated equally in reliability, the difference between advice/observation might not be so strong.

Minor things:

Line 90 and 125: Wasn't ever made clear why the analysis happened for the 5th trial only (and 6 - 15 after that). Were the first 4 trials asocial learning only? This is of course fine, but I don't think it was mentioned and some justification would be good.

Line 94 - I was left wondering what your main hypotheses were here, maybe say "detailed below in the methods section" to avoid making me think I have to check the pre-reg/ supp material to find them!

Line 111 - I'd like more payment detail here, how long did the study last, did this conform to minimum wage for those who got no bonus/ were the slowest? I don't really want to have to check supplementary material to know this, I think we should always say every participant achieved living wage standards as default in methods sections to make this the norm. Also the supplementary material doesn't actually tell us about participant payment/rewards/experiment length.

Line 116: Could you say how the experiment was coded? Was it Gorilla, O-tree, something else? It would be useful for future researchers to know, if they want to use the same task, can they, is the code available?

Line 128- Again this is not my area but it's not clear anywhere why fluid intelligence was given? It's not mentioned in the pre-reg, what was the justification for this?

Line 156: It's confusing reading your Hypotheses and them not being identical to the pre-reg ones. Rather than the mini paragraph at the end saying "also paranoia, H1a, H2a etc" I think it makes more sense to the reader to integrate these into the hypotheses, and keep it identical to the pre-reg. Also as the paranoia aspect is the novel/interesting part of this, and the main point of interest, it's worth mentioning straight away, not briefly at the end.

Pre-reg discrepancy: You mention Cumulative link models in the pre-reg but don't appear to in the main analysis, just a quick mention of why that happened/what the justification was for not using these models would be good.

Typos/clarity:

I don't quite understand the sentence starting line 204, or how this supports the pre-registered hypothesis about assimilation... also is line 202 a typo, that's a lot of zeros on the p-value?

Typo line 435 " these" differences

Review form: Reviewer 2

Recommendation

Accept with minor revision (please list in comments)

Scientific importance: Is the manuscript an original and important contribution to its field?

Good

General interest: Is the paper of sufficient general interest?

Good

Quality of the paper: Is the overall quality of the paper suitable?

Excellent

Is the length of the paper justified?

Yes

Should the paper be seen by a specialist statistical reviewer?

No

Do you have any concerns about statistical analyses in this paper? If so, please specify them explicitly in your report.

No

It is a condition of publication that authors make their supporting data, code and materials available - either as supplementary material or hosted in an external repository. Please rate, if applicable, the supporting data on the following criteria.

Is it accessible?

Yes

Is it clear?

Yes

Is it adequate?

Yes

Do you have any ethical concerns with this paper?

No

Comments to the Author

The manuscript "Trusting and learning from others: immediate and longterm effects of learning from observation and advice" reports a pre-registered, online experiment that investigates social and individual learning dynamics after both passive observation and advice.

Overall, this paper was a pleasure to read. It is very clearly structured and well written. All predictions and analysis plans were preregistered and the authors clearly distinguish between preregistered and exploratory analyses throughout the manuscript. I could also download all relevant data and code from OSF. One particular strength of the paper lies in the thorough analysis of behavioral results combined with simulation and computational modeling of learning dynamics. It is great to see how the authors first show empirical patterns, then simulate from different generative processes to show the data they would imply and, finally, fit the computational models to the data.

However, there are also some concerns about the experimental manipulation, the theoretical background and the implementation of the computational models.

The authors systematically varied (within-subjects) the framing of social information as observation and advice and hypothesized that individuals should pay more attention to advice because possible reputation effects make it more likely that advisors provide adaptive information. To validate this interpretation of group differences, the authors should have included some manipulation check to ensure that the higher perceived informational value is really what is driving the effect. A plausible alternative explanation would be that participants simply followed the normative information implied in the primes: “Secretly observe” very much sounds like cheating, whereas “receive advice” might generate a normative expectation to follow the given advice. Ideally, the difference between observation and advice should have emerged from the structure of the experiment, instead of being explicitly told to participants. This is particularly relevant as the evolutionary logic of social learning strategies does not depend on explicit representations, but likely responds to simpler cues in the (social) environment. The fact that paranoia modulates the effect is not sufficient evidence, especially as self-rated paranoia might itself be influenced by the behavior in the experiment. It might be plausible that participants who did not follow the advice rated themselves as higher in paranoia, simply because they have recently experienced themselves as not trusting advice. I do not think these concerns necessarily invalidate the drawn conclusions but the authors should thoroughly discuss possible interpretations of the empirical patterns.

I was also wondering why the authors decided to make social information always adaptive, as this makes it very difficult to disentangle individual and social information use (especially because social information is provided just once and not every round as in other experiments, e.g. Toyokawa et al., 2019; Deffner et al., 2020). Social information comes only after 4 rounds, when individuals already begin to learn which option is optimal. Following social information then is likely to produce a large payoff, which further increases individuals’ preference. Providing non-adaptive information could create a nice contrast between the influence of observation/advice and personally experienced payoffs.

The social learning literature discusses phenomena very similar to what the authors call “advice” using the term “teaching”. There are multiple theoretical models (e.g. Castro & Toro, 2014; Fogarty et al., 2011) that explore the evolutionary conditions under which tutors should give a pupil adaptive advice even if it comes with some cost to themselves. The introduction would benefit from some discussion about how advice relates to teaching and which evolutionary dynamics might underpin both.

Relatedly, in the introduction, the authors should better justify why they included a measure of paranoia and not of any other personality trait that might modulate advice taking. At the moment, it appears rather ad-hoc and should be better embedded in theory.

The authors used simple reinforcement learning models and estimated individual-level parameters that modulate the value and inverse temperature after receiving social information. They fitted a single parameter per participant for rounds 6-15 which might hide more intricate learning dynamics. The bimodal distribution of the beta boost, for instance, might not reflect the real distribution of inter-individual variation, but might be the result of learning trajectories that change differently over time. In addition, it might be constructive to compare how learning parameters differ before and after social information is provided. Researchers have implemented time-varying learning parameters in similar models that allow a closer look at the temporal dynamics of learning strategies (e.g. Deffner et al., 2020).

The authors should also provide more information on how the computational model was fitted. Based on the code, it seems like the authors used some custom made function instead of relying on established statistical inference framework. This is of course ok, but in this case, it is even

more important to provide enough details on the exact procedure.

Minor points:

Line 141: "Independent variables" instead of "dependent variables"

Literature cited:

Castro, L., & Toro, M. A. (2014). Cumulative cultural evolution: the role of teaching. *Journal of Theoretical Biology*, 347, 74-83.

Deffner, D., Kleinow, V., & McElreath, R. (2020). Dynamic social learning in temporally and spatially variable environments. *Royal Society open science*, 7(12), 200734.

Fogarty, L., Strimling, P., & Laland, K. N. (2011). The evolution of teaching. *Evolution: International Journal of Organic Evolution*, 65(10), 2760-2770.

Toyokawa, W., Whalen, A., & Laland, K. N. (2019). Social learning strategies regulate the wisdom and madness of interactive crowds. *Nature Human Behaviour*, 3(2), 183-193.

Decision letter (RSPB-2021-1414.R0)

23-Jul-2021

Dear Dr Hertz:

I have now received input from two reviewers and the associate editor on your manuscript. Both they and I find your manuscript to be well written and interesting. However, the reviewers have raised several issues, and based on their advice as well as my own read of your manuscript, I am asking you to revise it to take these considerations into account. Their comments can be found appended at the end of this email, and these are thoughtful and thorough, so I will not repeat them here. However, I encourage you to pay particular attention to their comments regarding your choice to make the advice/observation always accurate, how the players interpreted the advice/observation (and whether one or both may have been interpreted differently than intended), whether learning in the first 4 trials influenced attention to the advice/observation, the role of trust in paranoia, and providing additional detail on the computational model.

Your manuscript has now been peer reviewed and the reviews have been assessed by an Associate Editor. The reviewers' comments (not including confidential comments to the Editor) and the comments from the Associate Editor are included at the end of this email for your reference. As you will see, the reviewers and the Editors have raised some concerns with your manuscript and we would like to invite you to revise your manuscript to address them.

When submitting your revision please upload a file under "Response to Referees" - in the "File Upload" section. This should document, point by point, how you have responded to the reviewers' and Editors' comments, and the adjustments you have made to the manuscript. We

require a copy of the manuscript with revisions made since the previous version marked as 'tracked changes' to be included in the 'response to referees' document.

Research ethics:

Use of animals and field studies:

It is a condition of publication that you make available the data and research materials supporting the results in the article. Please see our Data Sharing Policies (<https://royalsociety.org/journals/authors/author-guidelines/#data>). Datasets should be deposited in an appropriate publicly available repository and details of the associated accession number, link or DOI to the datasets must be included in the Data Accessibility section of the article (<https://royalsociety.org/journals/ethics-policies/data-sharing-mining/>). Reference(s) to datasets should also be included in the reference list of the article with DOIs (where available).

Online supplementary material will also carry the title and description provided during submission, so please ensure these are accurate and informative. Note that the Royal Society will not edit or typeset supplementary material and it will be hosted as provided. Please ensure that

the supplementary material includes the paper details (authors, title, journal name, article DOI). Your article DOI will be 10.1098/rspb.[paper ID in form xxxx.xxxx e.g. 10.1098/rspb.2016.0049].

Please submit a copy of your revised paper within three weeks. If we do not hear from you within this time your manuscript will be rejected. If you are unable to meet this deadline please let us know as soon as possible, as we may be able to grant a short extension.

Best wishes,
Dr Sarah Brosnan
Editor, Proceedings B
mailto: proceedingsb@royalsociety.org

Associate Editor

Comments to Author:

Two reviewers have provided feedback on this article and both praised the clarity of presentation as well as the novelty of the question addressed. However, both reviewers also suggest a number of ways in which the reporting can be enhanced and clarified.

Reviewer(s)' Comments to Author:

Referee: 1

Comments to the Author(s)

I really enjoyed this manuscript. I commend the authors' dedication to open and transparent research practices; pre-registering their hypotheses and making all data and analyses openly available. All papers are easier to review when this is the case.

I think the study is very interesting, combining paranoia with social learning tendencies is definitely a gap in the literature and an important avenue for exploring individual differences in cognition and how they impact social learning. Also the use of "advice" versus "observation" is definitely underexplored and worth more investigation.

I think the task is also really fun and engaging for participants. And beautiful, clear figures.

I like the combination of experiment and model also, although I have to say here that I am awful at understanding models, this is a big gap in my knowledge. So I am not going to comment on the modelling, please find someone else to do that. There are plenty of other social learning modellers out there who would be much better at examining this than me (E.g. Elena Miu, Tom Morgan, Bret Beheim, Paul Smaldino, Anne Kandler...).

Mainly I just have a few conceptual clarifications and extra details to strengthen the manuscript.

Conceptual clarifications:

Prestige v reputation:

I like your discussion of reputation as a mechanism for replacing a more cognitively demanding evaluation of success. This fits very nicely with recent work on prestige bias, and I wonder whether you see reputation as a kind of prestige, and if so, it is worth mentioning that I think. I think it would be worth engaging with the more recent prestige literature than the cited Atkisson 2012 paper. E.g.

Brand, C. O., Mesoudi, A., & Morgan, T. (2021). Trusting the experts: the domain-specificity of prestige-biased social learning.

Brand, C. O., Heap, S., Morgan, T. J. H., & Mesoudi, A. (2020). The emergence and adaptive use of prestige in an online social learning task. *Scientific reports*, 10(1), 1-11.

Jiménez, Á. V., & Mesoudi, A. (2019). Prestige-biased social learning: current evidence and outstanding questions. *Palgrave Communications*, 5(1), 1-12.

Cheng, J. T., Tracy, J. L., Foulsham, T., Kingstone, A., & Henrich, J. (2013). Two ways to the top: evidence that dominance and prestige are distinct yet viable avenues to social rank and influence. *Journal of personality and social psychology*, 104(1), 103.

For example, in light of the Scientific Reports paper above, it can't be the case that reputation is necessary to "obviate the need of receivers to evaluate the demonstrator's abilities or expertise" (line 378), as this is achieved by prestige-bias; instead of having to evaluate expertise/success yourself, you can use prestige cues that have emerged via the social learning dynamics themselves. Unless you would view someone who is "most copied/paid attention" as a type of reputation in the way you discuss it? Maybe this is in fact what you mean, either way I think it would help to more explicitly mention how you think reputation does/doesn't differ from prestige, and reference the interplay between prestige and advice-giving.

Advice:

Similarly, it's not made entirely explicit or clear in the introduction / hypotheses why you think advice will be used more than observation, you have to sort of read between the lines.

Related, I like the penultimate discussion paragraph about your own evaluation of information versus another's evaluation. I think it's worth mentioning though (maybe in this paragraph or elsewhere) that if someone is giving advice, it is presumably implied/inferred that this advice is an amalgamation of (usually many) different experiences. It would be odd for someone to give advice based on just one dip in the lake, whereas in the observation condition you are just viewing one dip in the lake. It's possible that participants assumed that the advice-giver had finished their session of lake dipping, and based on all their dips, they are advising this lake. I know participants were never explicitly told that the advice-giver had taken more than one dip, but what were they told? And what might they have inferred? It would be nice to ask them post-experiment, I know it's too late for that now so maybe just mention that one possible explanation is that once information is given in the form of advice, it may be assumed that it is made up of more experiences than just one.

Trust :

Similarly in the discussion you mention "trust" a lot in the abstract and again in the first discussion paragraph, but you didn't actually measure trust in the demonstrators, right? Or are you assuming trust to be wrapped-up in the paranoia spectrum? Isn't it possible that two people on the same place on the paranoia spectrum might have different levels of trust in one particular model for different reasons? E.g. two people might be high in paranoia, but for some reason they have different levels of trust in Adviser A compared to Adviser B? Please elaborate on this.

Eavesdropping and bad intentions:

I follow the logic of why a paranoid person may trust what someone says *to them* less than what they see that person do "secretly" or through "eavesdropping," but I'm not entirely convinced. Particularly line 70-71 in the intro, and mentioned again in the discussion, I'm not entirely convinced a paranoid person would trust someone if they were eavesdropping more than when they were being told explicitly. Surely if I am paranoid and I believe people have bad intentions, then a person is just as likely to be lying to someone else (while I secretly listen to them) as they are likely to be lying to me when they directly address me? But I am no specialist in

paranoia so please correct me if I've misunderstood (or elaborate on this slightly in the introduction for people outside the area).

Reliability of social info:

Line 88 - Why was social information always reliable? You mention to avoid differences between the two social learning conditions, but surely they could both vary in reliability to the same degree? It wasn't mentioned in the pre-reg either so it just seems a bit odd to make social learning unrealistically reliable. But maybe it was simpler, and you can't change it now, so perhaps just mention that this is of course unrealistic but was for simplicity, and mention this as a limitation in the discussion. E.g. If you knew someone's advice was always 100% accurate - of course you'd take it! I know the interesting bit then is then why advice is preferred over observation, but it's possible that if they both fluctuated equally in reliability, the difference between advice/observation might not be so strong.

Minor things:

Line 90 and 125: Wasn't ever made clear why the analysis happened for the 5th trial only (and 6 - 15 after that). Were the first 4 trials asocial learning only? This is of course fine, but I don't think it was mentioned and some justification would be good.

Line 94 - I was left wondering what your main hypotheses were here, maybe say "detailed below in the methods section" to avoid making me think I have to check the pre-reg/ supp material to find them!

Line 111 - I'd like more payment detail here, how long did the study last, did this conform to minimum wage for those who got no bonus/ were the slowest? I don't really want to have to check supplementary material to know this, I think we should always say every participant achieved living wage standards as default in methods sections to make this the norm. Also the supplementary material doesn't actually tell us about participant payment/rewards/experiment length.

Line 116: Could you say how the experiment was coded? Was it Gorilla, O-tree, something else? It would be useful for future researchers to know, if they want to use the same task, can they, is the code available?

Line 128- Again this is not my area but it's not clear anywhere why fluid intelligence was given? It's not mentioned in the pre-reg, what was the justification for this?

Line 156: It's confusing reading your Hypotheses and them not being identical to the pre-reg ones. Rather than the mini paragraph at the end saying "also paranoia, H1a, H2a etc" I think it makes more sense to the reader to integrate these into the hypotheses, and keep it identical to the pre-reg. Also as the paranoia aspect is the novel/interesting part of this, and the main point of interest, it's worth mentioning straight away, not briefly at the end.

Pre-reg discrepancy: You mention Cumulative link models in the pre-reg but don't appear to in the main analysis, just a quick mention of why that happened/what the justification was for not using these models would be good.

Typos/clarity:

I don't quite understand the sentence starting line 204, or how this supports the pre-registered hypothesis about assimilation... also is line 202 a typo, that's a lot of zeros on the p-value?

Typo line 435 " these" differences

Referee: 2

Comments to the Author(s)

The manuscript “Trusting and learning from others: immediate and longterm effects of learning from observation and advice” reports a pre-registered, online experiment that investigates social and individual learning dynamics after both passive observation and advice.

Overall, this paper was a pleasure to read. It is very clearly structured and well written. All predictions and analysis plans were preregistered and the authors clearly distinguish between preregistered and exploratory analyses throughout the manuscript. I could also download all relevant data and code from OSF. One particular strength of the paper lies in the thorough analysis of behavioral results combined with simulation and computational modeling of learning dynamics. It is great to see how the authors first show empirical patterns, then simulate from different generative processes to show the data they would imply and, finally, fit the computational models to the data.

However, there are also some concerns about the experimental manipulation, the theoretical background and the implementation of the computational models.

The authors systematically varied (within-subjects) the framing of social information as observation and advice and hypothesized that individuals should pay more attention to advice because possible reputation effects make it more likely that advisors provide adaptive information. To validate this interpretation of group differences, the authors should have included some manipulation check to ensure that the higher perceived informational value is really what is driving the effect. A plausible alternative explanation would be that participants simply followed the normative information implied in the primes: “Secretly observe” very much sounds like cheating, whereas “receive advice” might generate a normative expectation to follow the given advice. Ideally, the difference between observation and advice should have emerged from the structure of the experiment, instead of being explicitly told to participants. This is particularly relevant as the evolutionary logic of social learning strategies does not depend on explicit representations, but likely responds to simpler cues in the (social) environment. The fact that paranoia modulates the effect is not sufficient evidence, especially as self-rated paranoia might itself be influenced by the behavior in the experiment. It might be plausible that participants who did not follow the advice rated themselves as higher in paranoia, simply because they have recently experienced themselves as not trusting advice. I do not think these concerns necessarily invalidate the drawn conclusions but the authors should thoroughly discuss possible interpretations of the empirical patterns.

I was also wondering why the authors decided to make social information always adaptive, as this makes it very difficult to disentangle individual and social information use (especially because social information is provided just once and not every round as in other experiments, e.g. Toyokawa et al., 2019; Deffner et al., 2020). Social information comes only after 4 rounds, when individuals already begin to learn which option is optimal. Following social information then is likely to produce a large payoff, which further increases individuals’ preference. Providing non-adaptive information could create a nice contrast between the influence of observation/advice and personally experienced payoffs.

The social learning literature discusses phenomena very similar to what the authors call “advice” using the term “teaching”. There are multiple theoretical models (e.g. Castro & Toro, 2014; Fogarty et al., 2011) that explore the evolutionary conditions under which tutors should give a pupil adaptive advice even if it comes with some cost to themselves. The introduction would benefit from some discussion about how advice relates to teaching and which evolutionary dynamics might underpin both.

Relatedly, in the introduction, the authors should better justify why they included a measure of paranoia and not of any other personality trait that might modulate advice taking. At the moment, it appears rather ad-hoc and should be better embedded in theory.

The authors used simple reinforcement learning models and estimated individual-level parameters that modulate the value and inverse temperature after receiving social information. They fitted a single parameter per participant for rounds 6-15 which might hide more intricate learning dynamics. The bimodal distribution of the beta boost, for instance, might not reflect the real distribution of inter-individual variation, but might be the result of learning trajectories that change differently over time. In addition, it might be constructive to compare how learning parameters differ before and after social information is provided. Researchers have implemented time-varying learning parameters in similar models that allow a closer look at the temporal dynamics of learning strategies (e.g. Deffner et al., 2020).

The authors should also provide more information on how the computational model was fitted. Based on the code, it seems like the authors used some custom made function instead of relying on established statistical inference framework. This is of course ok, but in this case, it is even more important to provide enough details on the exact procedure.

Minor points:

Line 141: "Independent variables" instead of "dependent variables"

Literature cited:

Castro, L., & Toro, M. A. (2014). Cumulative cultural evolution: the role of teaching. *Journal of Theoretical Biology*, 347, 74-83.

Deffner, D., Kleinow, V., & McElreath, R. (2020). Dynamic social learning in temporally and spatially variable environments. *Royal Society open science*, 7(12), 200734.

Fogarty, L., Strimling, P., & Laland, K. N. (2011). The evolution of teaching. *Evolution: International Journal of Organic Evolution*, 65(10), 2760-2770.

Toyokawa, W., Whalen, A., & Laland, K. N. (2019). Social learning strategies regulate the wisdom and madness of interactive crowds. *Nature Human Behaviour*, 3(2), 183-193.

Author's Response to Decision Letter for (RSPB-2021-1414.R0)

See Appendix A.

RSPB-2021-1414.R1 (Revision)

Review form: Reviewer 1

Recommendation

Accept with minor revision (please list in comments)

Scientific importance: Is the manuscript an original and important contribution to its field?

Good

General interest: Is the paper of sufficient general interest?

Good

Quality of the paper: Is the overall quality of the paper suitable?

Excellent

Is the length of the paper justified?

Yes

Should the paper be seen by a specialist statistical reviewer?

No

Do you have any concerns about statistical analyses in this paper? If so, please specify them explicitly in your report.

No

It is a condition of publication that authors make their supporting data, code and materials available - either as supplementary material or hosted in an external repository. Please rate, if applicable, the supporting data on the following criteria.

Is it accessible?

Yes

Is it clear?

Yes

Is it adequate?

Yes

Do you have any ethical concerns with this paper?

No

Comments to the Author

Thank you for all your clarifications and responses. A few minor issues remain:

In response to my point about advice being assumed to be an amalgamation of experiences, compared to a single observation, you say : " In the instructions, participants were informed that they were either receiving advice from or secretly observing the choice of an expert player. We used this language specifically to address the important confound you raised above. "

But it is not clear to me how that language does address the important confound I raise. It could be the case that they either receive advice (made up of a number of experiences) from an expert player, or observe just one experience of an expert player. For this reason, and to get a sense of what the participants may have assumed/understood from the instructions, I think it's necessary to include a screenshot of the exact wording and instructions that the participants saw, either in the main manuscript, or in the supplementary material. This is particularly useful for researchers if they are wanting to replicate or base their own experiments on your work in the future.

The final point: you mention after the participant payment information "No participant was excluded from analysis, but single trials that took more than 20 seconds to complete were removed." This is not mentioned anywhere in the preregistration and I wonder what it was about 20 seconds that determined those trials were removed. A simple justifying statement would suffice, and ideally these data exclusion criteria are included in future preregistrations. e.g. can you confirm that your results wouldn't change if the cut off was 25 seconds, or 15 seconds, or no cut-off at all?

Review form: Reviewer 2

Recommendation

Accept as is

Scientific importance: Is the manuscript an original and important contribution to its field?

Good

General interest: Is the paper of sufficient general interest?

Good

Quality of the paper: Is the overall quality of the paper suitable?

Excellent

Is the length of the paper justified?

Yes

Should the paper be seen by a specialist statistical reviewer?

No

Do you have any concerns about statistical analyses in this paper? If so, please specify them explicitly in your report.

No

It is a condition of publication that authors make their supporting data, code and materials available - either as supplementary material or hosted in an external repository. Please rate, if applicable, the supporting data on the following criteria.

Is it accessible?

Yes

Is it clear?

Yes

Is it adequate?

Yes

Do you have any ethical concerns with this paper?

No

Comments to the Author

Thanks for the very thorough and clear revisions that addressed all of the comments I had! I do not doubt the validity of the computational analyses in any way, but I would recommend the authors to use one of the standard probabilistic programming frameworks (such as stan or PyMC3) to fit such computational models in the future, instead of their custom, in-house code. This would make the models more transparent to others and would also increase the chances that other researchers will build on this great work. I am excited to see this published soon!

Decision letter (RSPB-2021-1414.R1)

27-Sep-2021

Dear Dr Hertz

I am pleased to inform you that your manuscript RSPB-2021-1414.R1 entitled "Trusting and learning from others: immediate and long-term effects of learning from observation and advice" has been accepted for publication in Proceedings B pending some minor revisions. As you will see below, Reviewer 2 requests two final points, both of which I think will add to the manuscript. Therefore, I invite you to respond to the referee(s)' comments and revise your manuscript. Because the schedule for publication is very tight, it is a condition of publication that you submit the revised version of your manuscript within 7 days. If you do not think you will be able to meet this date please let us know.

In order to ensure effective and robust dissemination and appropriate credit to authors the dataset(s) used should be fully cited. To ensure archived data are available to readers, authors should include a 'data accessibility' section immediately after the acknowledgements section.

This should list the database and accession number for all data from the article that has been made publicly available, for instance:

[http://datadryad.org/submit?journalID=RSPB&manu=\(Document not available\)](http://datadryad.org/submit?journalID=RSPB&manu=(Document%20not%20available)) which will take you to your unique entry in the Dryad repository. If you have already submitted your data to dryad you can make any necessary revisions to your dataset by following the above link. Please see <https://royalsocietypublishing.org/journals/ethics-policies/data-sharing-mining/> for more details.

6) For more information on our Licence to Publish, Open Access, Cover images and Media summaries, please visit <https://royalsocietypublishing.org/journals/authors/author-guidelines/>.

Sincerely,
Dr Sarah Brosnan
Editor, Proceedings B
<mailto:proceedingsb@royalsocietypublishing.org>

Associate Editor:
Board Member: 1
Comments to Author:

Thank you very much for your careful and detailed revisions and responses to the reviewer feedback. Like the reviewers I agree that you have thoroughly addressed the reviewers' comments and questions. However, Reviewer 2 raises two points in reference to additional clarification regarding your methods and treatment of your data that I believe are worth addressing to aid understanding of your protocols and potential replications of your methods.

Reviewer(s)' Comments to Author:

Referee: 2

Comments to the Author(s)

Thanks for the very thorough and clear revisions that addressed all of the comments I had! I do not doubt the validity of the computational analyses in any way, but I would recommend the authors to use one of the standard probabilistic programming frameworks (such as Stan or PyMC3) to fit such computational models in the future, instead of their custom, in-house code. This would make the models more transparent to others and would also increase the chances that other researchers will build on this great work. I am excited to see this published soon!

Referee: 1

Comments to the Author(s)

Thank you for all your clarifications and responses. A few minor issues remain:

In response to my point about advice being assumed to be an amalgamation of experiences, compared to a single observation, you say : " In the instructions, participants were informed that they were either receiving advice from or secretly observing the choice of an expert player. We used this language specifically to address the important confound you raised above. "

But it is not clear to me how that language does address the important confound I raise. It could be the case that they either receive advice (made up of a number of experiences) from an expert player, or observe just one experience of an expert player. For this reason, and to get a sense of what the participants may have assumed/understood from the instructions, I think it's necessary to include a screenshot of the exact wording and instructions that the participants saw, either in the main manuscript, or in the supplementary material. This is particularly useful for researchers if they are wanting to replicate or base their own experiments on your work in the future.

The final point: you mention after the participant payment information "No participant was excluded from analysis, but single trials that took more than 20 seconds to complete were removed." This is not mentioned anywhere in the preregistration and I wonder what it was about 20 seconds that determined those trials were removed. A simple justifying statement would suffice, and ideally these data exclusion criteria are included in future preregistrations. e.g. can you confirm that your results wouldn't change if the cut off was 25 seconds, or 15 seconds, or no cut-off at all?

Author's Response to Decision Letter for (RSPB-2021-1414.R1)

See Appendix B.

Decision letter (RSPB-2021-1414.R2)

29-Sep-2021

Dear Dr Hertz

I am pleased to inform you that your manuscript entitled "Trusting and learning from others: immediate and long-term effects of learning from observation and advice" has been accepted for publication in Proceedings B.

Data Accessibility section

Open Access

Paper charges

Sincerely,

Proceedings B

Appendix A

Dear Prof. Brosnan

Thank you for the helpful comments, and the opportunity to revise our manuscript.

We revised the manuscript to address the way participants perceived the task, and discuss alternative explanations and interpretations. We also examine the effect of participant's initial experience with the task (i.e. in the first 4 trials) on social learning, especially when this experience goes against the social information provided to the participants. We show that while social information was always correct, from the point of view of the participant this is not always the case.

Point-by-point responses are provided below.

At the end of this document we attached the revised manuscript, with changes highlighted.

Our replies to the reviewers' comments are provided in red. Any copied sections from the manuscript are given in blue.

Best wishes,

Uri Hertz, Vaughan Bell and Nichola Raihani

Reviewer(s)' Comments to Author:

Referee: 1

Comments to the Author(s)

I really enjoyed this manuscript. I commend the authors' dedication to open and transparent research practices; pre-registering their hypotheses and making all data and analyses openly available. All papers are easier to review when this is the case.

I think the study is very interesting, combining paranoia with social learning tendencies is definitely a gap in the literature and an important avenue for exploring individual differences in cognition and how they impact social learning. Also the use of "advice" versus "observation" is definitely underexplored and worth more investigation.

I think the task is also really fun and engaging for participants. And beautiful, clear figures.

I like the combination of experiment and model also, although I have to say here that I am awful at understanding models, this is a big gap in my knowledge. So I am not going to comment on the modelling, please find someone else to do that. There are plenty of other social learning modellers out there who would be much better at examining this than me (E.g. Elena Miu, Tom Morgan, Bret Beheim, Paul Smaldino, Anne Kandler...).

Mainly I just have a few conceptual clarifications and extra details to strengthen the manuscript.

- Thank you for the positive comments on our manuscript as well as the helpful suggestions.

Conceptual clarifications:

Prestige v reputation:

I like your discussion of reputation as a mechanism for replacing a more cognitively demanding evaluation of success. This fits very nicely with recent work on prestige bias, and I wonder whether you see reputation as a kind of prestige, and if so, it is worth mentioning that I think. I think it would be worth engaging with the more recent prestige literature than the cited Atkisson 2012 paper. E.g.

Brand, C. O., Mesoudi, A., & Morgan, T. (2021). Trusting the experts: the domain-specificity of prestige-biased social learning.

Brand, C. O., Heap, S., Morgan, T. J. H., & Mesoudi, A. (2020). The emergence and adaptive use of prestige in an online social learning task. *Scientific reports*, 10(1), 1-11.

Jiménez, Á. V., & Mesoudi, A. (2019). Prestige-biased social learning: current evidence

and outstanding questions. Palgrave Communications, 5(1), 1-12.

Cheng, J. T., Tracy, J. L., Foulsham, T., Kingstone, A., & Henrich, J. (2013). Two ways to the top: evidence that dominance and prestige are distinct yet viable avenues to social rank and influence. *Journal of personality and social psychology*, 104(1), 103.

For example, in light of the Scientific Reports paper above, it can't be the case that reputation is necessary to "obviate the need of receivers to evaluate the demonstrator's abilities or expertise" (line 378), as this is achieved by prestige-bias; instead of having to evaluate expertise/success yourself, you can use prestige cues that have emerged via the social learning dynamics themselves. Unless you would view someone who is "most copied/paid attention" as a type of reputation in the way you discuss it? Maybe this is in fact what you mean, either way I think it would help to more explicitly mention how you think reputation does/doesn't differ from prestige, and reference the interplay between prestige and advice-giving.

- Thanks for this comment. We agree that we could have been more exact with the terminology for reputation, prestige and status. However, rather than viewing reputation as a kind of prestige, in line with Cheng, we see prestige as one kind of reputation and having a prestigious reputation can increase status. We have amended this to be clearer in the Introduction of the manuscript (see below, lines 54-55) and now cite the papers you mentioned as well.

[16–18]. Advisers who provide accurate information which benefits others might appear more knowledgeable, resulting in greater influence in the future and enhanced prestige [13,15,19]. On the other hand, making a mistake or giving the wrong advice can negatively impact an adviser's status and down-weight the probability that they will be copied by others [13,20–23]. The risk of losing prestige can affect advice-giving strategies: for example advisers might offer advice only when they believe that their information is accurate [14,17,24].

Advice:

Similarly, it's not made entirely explicit or clear in the introduction / hypotheses why you think advice will be used more than observation, you have to sort of read between the lines.

- Paragraph 3 in the introduction deals with this hypothesis and we have edited it lightly to make it less ambiguous (see above).

Related, I like the penultimate discussion paragraph about your own evaluation of information versus another's evaluation. I think it's worth mentioning though (maybe in this paragraph or elsewhere) that if someone is giving advice, it is presumably implied/inferred that this advice is an amalgamation of (usually many) different experiences. It would be odd for someone to give advice based on just one dip in the lake, whereas in the observation condition you are just viewing one dip in the lake. It's possible that participants assumed that the advice-giver had finished their session of lake dipping, and based on all their dips, they are advising this lake. I know participants were never explicitly told that the advice-giver had taken more than one dip, but what were they told? And what might they have inferred? It would be nice to ask them post-experiment, I know it's too late for that now so maybe just mention that one possible explanation is that once information is given in the form of advice, it may be assumed that it is made up of more experiences than just one.

- Thanks for this comment. In the instructions, participants were informed that they were either receiving advice from or secretly observing the choice of an expert player. We used this language specifically to address the important confound you raised above.

Trust :

Similarly in the discussion you mention "trust" a lot in the abstract and again in the first discussion paragraph, but you didn't actually measure trust in the demonstrators, right? Or are you assuming trust to be wrapped-up in the paranoia spectrum? Isn't it possible that two people on the same place on the paranoia spectrum might have different levels of trust in one particular model for different reasons? E.g. two people might be high in paranoia, but for some reason they have different levels of trust in Adviser A compared to Adviser B? Please elaborate on this.

- Apologies if this was unclear. Trust is a fundamental component of paranoia and we have used the term trust to mean whether a participant follows the advice of a demonstrator. Our expectation is that variation in trust will be explained by score on the GPTS (the paranoia instrument we used) but of course there is unlikely to be a 1:1 mapping of paranoia onto whether or not someone trusts (or follows the advice) of another person. We have clarified this in lines 67-72 of the introduction.

An exaggerated tendency to believe that others have malign intentions - and associated suspicion and mistrust - is the basis of paranoia [31,32], which is not solely a clinical category but also varies across a full spectrum of severity in the general population [31,33,34]. Given the increased tendency to believe that others have harmful intentions, even when true intentions are ambiguous [35,36], we might expect higher levels of paranoia to be associated with lower levels of trust in others and, consequently, a reduced tendency to follow advice.

Eavesdropping and bad intentions:

I follow the logic of why a paranoid person may trust what someone says *to them* less than what they see that person do “secretly” or through “eavesdropping,” but I’m not entirely convinced. Particularly line 70-71 in the intro, and mentioned again in the discussion, I’m not entirely convinced a paranoid person would trust someone if they were eavesdropping more than when they were being told explicitly. Surely if I am paranoid and I believe people have bad intentions, then a person is just as likely to be lying to someone else (while I secretly listen to them) as they are likely to be lying to me when they directly address me? But I am no specialist in paranoia so please correct me if I’ve misunderstood (or elaborate on this slightly in the introduction for people outside the area).

- We believe this comment stems from a misunderstanding of what the eavesdropping condition entailed. In the eavesdropping condition, participants did not believe they were witnessing a demonstrator *giving advice* to another participant; instead they were told that they were copying the demonstrator’s actual choice and that the demonstrator was unaware that they were being observed. This distinction is actually fundamental to our hypothesis as we don’t expect participants to assume that a person they are secretly observing choosing one lake or another will intentionally try to mislead them, or indeed, anyone else - and so this is why we expected to observe a reduced tendency to follow advice in paranoia, where these concerns about the intentions of demonstrators *are* relevant.

Reliability of social info:

Line 88 - Why was social information always reliable? You mention to avoid differences between the two social learning conditions, but surely they could both vary in reliability to the same degree? It wasn’t mentioned in the pre-reg either so it just seems a bit odd to make social learning unrealistically reliable. But maybe it was simpler, and you can’t change it now, so perhaps just mention that this is of course unrealistic but was for simplicity, and mention this as a limitation in the discussion. E.g. If you knew someone’s advice was always 100% accurate - of course you’d take it! I know the interesting bit then is then why advice is preferred over observation, but it’s possible that if they both

fluctuated equally in reliability, the difference between advice/observation might not be so strong.

- Thanks for this comment, which was also raised by Reviewer 2. It is of course true that social information always being reliable is unrealistic but we wanted to keep this feature of the experiment constant for tractability: what we were primarily interested in in this study was whether paranoia would differentially impact learning through advice compared to learning through observation. From the participants' point of view, there was no way to know that the social info was accurate. Providing accurate information allowed us to measure long-term compliance with the social information without the confound of information accuracy. It also allowed us to avoid deceiving our participants, whose earnings in the task depended on them choosing the best lake.
- Nevertheless, because there was high variance around the mean rewards offered by each lake, it was not always the case that the advised lake (i.e. the best lake) yielded more fish than the worse lake. We have conducted some further analysis, which shows that, for about 250 participants, out of 1200 who sampled both lakes in the first 4 trials, they experienced the good lake as less rewarding than the bad lake in the first 4 trials. For these participants, advice/observation was going against their initial experience (Figure R1-A).
- We found that social info was followed even when the good-lake was experienced as less profitable than the bad lake (Figure R1-B). However, advice was not significantly more followed than observation in these cases. We further examined the effect of initial experience on long term effects of social information, as was captured by the high-precision boost parameter in our model (Figure R1-C). Here we found no effect of initial experience on long-term effect.
- We added this figure, and statistical analyses of these effects, to the supplementary materials. We also now stress the fact that while social information was always accurate, this was not apparent to the participants, and indeed the social information went against their own experience with the lakes in some cases (L 91 – 94, 101-103, 130-132, 139-141 and supplementary materials).

Figure R1 - The effect of initial experience on immediate and long-term effects of social information. (A) Distributions of experienced difference in average rewards between good and bad lake. The dashed line indicates no difference. (B) Initial experience affected the likelihood to choose the good lake following social information, but did not change the main effects observed in Figure 2 of the main text. (C) Initial experience did not have effect on the likelihood of participants to demonstrate high beta-boost (precision boost). Bars indicate the means, and error-bars indicate 95% confidence intervals.

In all conditions, participants made choices without social information over the first four trials, to allow them to establish some expectations of the lakes. This was useful for the q-learning modeling (described below), which explored the effect of previous experience on advice taking.

...

Social information was always accurate (indicating the lake with the greater number of expected fish) to avoid introduction of irrelevant differences between social information

conditions, but was not always in line with the experience-based expectations of the participants, and did not always lead to immediate high rewards.

...

In the fishing task (Figure 1), participants ($N = 1492$) chose to fish at one of two lakes and subsequently received feedback about the number of fishes they caught. Participants made this choice 15 times for each pair of lakes. The number of fishes caught was randomly drawn from a normal distribution, with one lake having higher average yield than the other ($M_{\text{Good}}=5.5$, $M_{\text{Bad}}=4$), and both lakes having the same variance ($\sigma=1.7$). As these distributions overlapped, it was possible to receive high rewards from the bad lake and vice versa in some trials. At the end of the task, participants were paid a bonus that was determined by the total number of fishes they caught (20 pence per 50 fishes caught) (Average response time and reward for each block are in S10). The experiment was programmed in JavaScript, and the code is available in the open materials.

Each participant completed three blocks: control, observation and advice (order counter-balanced), with each block differing in the social information provided to participants after the 4th trial. Social information always recommended the good (high expected reward) lake. However, this was not immediately apparent to the participants, as their initial experience in the first 4 trials could go against the social information (see analysis of initial experience in the Supplementary Materials Figure S1).

Minor things:

Line 90 and 125: Wasn't ever made clear why the analysis happened for the 5th trial only (and 6 - 15 after that). Were the first 4 trials asocial learning only? This is of course fine, but I don't think it was mentioned and some justification would be good.

- A: We let participants experience the reward for a short while, allowing them to establish some expectations of the lakes. This was useful for the q-learning modeling, which explicitly looks at the effect of previous experience on advice taking. We now make this clear in lines 91-94 of the introduction (see below).

In all conditions, participants made choices without social information over the first four trials, to allow them to establish some expectations of the lakes. This was useful for the q-learning modeling (described below), which explored the effect of previous experience on advice taking. In the observation and advice conditions, social

information was presented once, after four trials and always suggested the higher-paying option.

Line 94 - I was left wondering what your main hypotheses were here, maybe say "detailed below in the methods section" to avoid making me think I have to check the pre-reg/ supp material to find them!

- We have included this clarifying statement.

Line 111 - I'd like more payment detail here, how long did the study last, did this conform to minimum wage for those who got no bonus/ were the slowest? I don't really want to have to check supplementary material to know this, I think we should always say every participant achieved living wage standards as default in methods sections to make this the norm. Also the supplementary material doesn't actually tell us about participant payment/rewards/experiment length.

- We have added a section on participant earnings into the methods section of the manuscript (lines 119 to 121).

All participants provided informed consent and received monetary compensation at a fixed rate of £1.50 GBP for participation, and could gain up to £1.80 as a performance-based bonus (average bonus = £0.91). On average, the task took 17 minutes to complete and participant earnings were equivalent to £8.22 per hour in the task. All participants received a bonus (minimum £0.80).

Line 116: Could you say how the experiment was coded? Was it Gorilla, O-tree, something else? It would be useful for future researchers to know, if they want to use the same task, can they, is the code available?

- The experiment was coded in-house, using JavaScript, and the script is provided in the online materials. We added the experiment code to the online materials (Line 134).

Line 128- Again this is not my area but it's not clear anywhere why fluid intelligence was given? It's not mentioned in the pre-reg, what was the justification for this?

- We did mention that we would use 'cognitive score' (measured using Hagen Matrices Test) in the pre-registration document, in the analyses section. This is

important to ensure that any differences found are not simply due to potential differences in cognitive ability between groups.

Line 156: It's confusing reading your Hypotheses and them not being identical to the pre-reg ones. Rather than the mini paragraph at the end saying "also paranoia, H1a, H2a etc" I think it makes more sense to the reader to integrate these into the hypotheses, and keep it identical to the pre-reg. Also as the paranoia aspect is the novel/interesting part of this, and the main point of interest, it's worth mentioning straight away, not briefly at the end.

- Thanks for this comment. This is a complex study and we attempted to write the manuscript in exactly the format you describe initially, but decided against it, as it was harder to follow than the current version. Having tried both structures, we would like to leave it in the current format. Since both reviewers and the editor have commented on the clarity of the writing, we hope that our decision to leave the manuscript structure will be acceptable.

Pre-reg discrepancy: You mention Cumulative link models in the pre-reg but don't appear to in the main analysis, just a quick mention of why that happened/what the justification was for not using these models would be good.

- Thank you for this point. This is of course, important, and one we covered in Section 8, point 2 of the pre-registration document, where we explain that we will use more traditional and widely-understood GLMM approach if our data conform to the assumptions underpinning these models, so we hope we have stayed consistent with our analysis plan during this aspect of the analysis.

Typos/clarity:

I don't quite understand the sentence starting line 204, or how this supports the pre-registered hypothesis about assimilation... also is line 202 a typo, that's a lot of zeros on the p-value?

- We have reworded the sentence to make it clear that the reduced time interval suggests that participants deliberated less when following advice compared to when they were copying observed choices (now line 222).
- The p value is correct, but the presentation does not comply with the style of displaying p values in the paper, which was to indicate any p lower than 0.001 as $p < 0.001$. We corrected it to comply with the presentation of statistics throughout the rest of the paper.

This suggests that participants deliberated less when following social information framed as advice compared to when using social information framed as arriving via observation.

Typo line 435 “ these” differences

- Many thanks for catching this. It is now corrected.

Referee: 2

Comments to the Author(s)

The manuscript “Trusting and learning from others: immediate and longterm effects of learning from observation and advice” reports a pre-registered, online experiment that investigates social and individual learning dynamics after both passive observation and advice.

Overall, this paper was a pleasure to read. It is very clearly structured and well written. All predictions and analysis plans were preregistered and the authors clearly distinguish between preregistered and exploratory analyses throughout the manuscript. I could also download all relevant data and code from OSF. One particular strength of the paper lies in the thorough analysis of behavioral results combined with simulation and computational modeling of learning dynamics. It is great to see how the authors first show empirical patterns, then simulate from different generative processes to show the data they would imply and, finally, fit the computational models to the data.

- Thank you for this positive feedback on our study.

However, there are also some concerns about the experimental manipulation, the theoretical background and the implementation of the computational models.

The authors systematically varied (within-subjects) the framing of social information as observation and advice and hypothesized that individuals should pay more attention to advice because possible reputation effects make it more likely that advisors provide adaptive information. To validate this interpretation of group differences, the authors should have included some manipulation check to ensure that the higher perceived informational value is really what is driving the effect. A plausible alternative explanation would be that participants simply followed the normative information implied in the primes: “Secretly observe” very much sounds like cheating, whereas “receive advice” might generate a normative expectation to follow the given advice. Ideally, the difference between observation and advice should have emerged from the structure of the

experiment, instead of being explicitly told to participants. This is particularly relevant as the evolutionary logic of social learning strategies does not depend on explicit representations, but likely responds to simpler cues in the (social) environment.

The fact that paranoia modulates the effect is not sufficient evidence, especially as self-rated paranoia might itself be influenced by the behavior in the experiment. It might be plausible that participants who did not follow the advice rated themselves as higher in paranoia, simply because they have recently experienced themselves as not trusting advice. I do not think these concerns necessarily invalidate the drawn conclusions but the authors should thoroughly discuss possible interpretations of the empirical patterns.

- We thank the reviewer for these additional thoughts. We agree that the setup of our experiment could be viewed as a bit contrived, but we don't believe that this is hugely different to many other experimental economics tasks, where (for example) participants are told that they can invest in a public good and that their choice will be observed by another player or that they can choose to trust or not trust someone based on the experimenter telling them how that person behaved in a previous round of a game. Many experimental economics tasks suffer from this similar problem of contrivance - and it is important to view such results (just like any results) not as the final word on a topic but as a brick in the wall of the evidence needed to come to a conclusion. We believe that the results of our experiment provide such a 'brick'. The use of the word 'secretly' was important in the context of our experiment as we wanted to ensure that, from a participant's point of view, there was no possibility that the demonstrator could be acting in a certain way because they were being observed. In other words, we wanted participants to believe that this was the lake the demonstrator genuinely preferred - this contrast allows us to then ask how participants treated this 'passive' social information with the more active social information that was transmitted as advice.
- On the point about people taking advice because they felt they ought to - and not because advice was seen as more informative, we acknowledge that this might account for part of the tendency to follow advice. Nevertheless, we agree with the reviewer that this does not represent a fatal problem for the inferences we draw as we were primarily interested in whether people were more likely to follow advice than observation - and whether this tendency would be reduced in paranoia (an effect for which we had concrete, a priori hypotheses).
- However, these are key point for consideration and we have included an explicit limitations section in the discussion where these issues are discussed further (Lines 461-484).

There are a number of limitations to be borne in mind when interpreting these data. First, the data all came from UK-based participants recruited via Prolific Academic, so it is not clear how these findings generalize to other groups and contexts. One of the limitations

of experimental designs in general, is that they are an engineered situation that may not fully reflect the complexity of the phenomenon as it occurs more broadly. In this respect, our study has much in common with other experimental economic games, where experimentally controlled social scenarios are used to glean insights about social preferences and behaviour. Here, the goal of experimental economics should be seen as understanding the directional effect of experimental factors on social preferences and behaviour, under the assumption that (*ceteris paribus*) the same direction of effects will be observed in the real world [76]. We also note that in our study, social information was always on-average informative, which does not reflect the potential range of accuracy of advice in broader social situations. Importantly though, the fact that social information was only statistically informative and did not always lead to immediate rewards, meant that this the informative nature of advice was likely not apparent to all participants due to the stochastic nature of the learning task (in other words, negative outcomes could occur after following advice). This decision was taken to allow for analytical tractability. Future work will explore how information reliability affects how people use (and continue to use) information received via advice versus observation. A final limitation is that we did not ask participants whether they perceived information gleaned via advice versus observation to be differentially reliable, or whether they felt obliged to follow the social information. Although participants were more likely to follow advice, some or all of this tendency could have been explained by a normative expectation that advice should be followed. Although our finding that advice-following was reduced in paranoia mitigates against this explanation, we cannot entirely rule it out.

I was also wondering why the authors decided to make social information always adaptive, as this makes it very difficult to disentangle individual and social information use (especially because social information is provided just once and not every round as in other experiments, e.g. Toyokawa et al., 2019; Deffner et al., 2020). Social information comes only after 4 rounds, when individuals already begin to learn which option is optimal. Following social information then is likely to produce a large payoff, which further increases individuals' preference. Providing non-adaptive information could create a nice contrast between the influence of observation/advice and personally experienced payoffs.

- As reviewer 1 also had a similar comment, and the answers to both reviewers are the same, we provide the same answer here. Please refer to the Reviewer 1 comment to view Figure R1.

- It is of course true that social information always being reliable is unrealistic but we wanted to keep this feature of the experiment constant for tractability: what we were primarily interested in in this study was whether paranoia would differentially impact learning through advice compared to learning through observation. From the participants' point of view, there was no way to know that the social info was accurate. Providing accurate information allowed us to measure long-term compliance with the social information without the confound of information accuracy, and it also allowed us to avoid deceiving our participants, whose earnings in the task depended on them choosing the best lake.
- Nevertheless, because there was high variance around the mean rewards offered by each lake, it was not always the case that the advised lake (i.e. the best lake) yielded more fish than the worse lake. We have conducted some further analysis, which shows that, for about 250 participants, out of 1200 who sampled both lakes in the first 4 trials, they experienced the good lake as less rewarding than the bad lake in the first 4 trials. For these participants, advice/observation was going against their initial experience (Figure R1-A).
- We found that social info was followed even when the good-lake was experienced as less profitable than the bad lake (Figure R1-B). However, advice was not significantly more followed than observation in these cases. We further examined the effect of initial experience on long term effects of social information, as was captured by the high-precision boost parameter in our model (Figure R1-C). Here we found no effect of initial experience on long-term effect.
- We added this figure, and statistical analyses of these effects, to the supplementary materials. We also now stress the fact that while social information was always accurate, this was not apparent to the participants, and indeed the social information went against their own experience with the lakes in some cases (L 91 – 94, 101-103, 130-132, 139-141 and supplementary materials).

Figure R1 - The effect of initial experience on immediate and long-term effects of social information. (A) Distributions of experienced difference in average rewards between good and bad lake. The dashed line indicates no difference. (B) Initial experience affected the likelihood to choose the good lake following social information, but did not change the main effects observed in Figure 2 of the main text. (C) Initial experience did not have effect on the likelihood of participants to demonstrate high beta-boost (precision boost). Bars indicate the means, and error-bars indicate 95% confidence intervals.

In all conditions, participants made choices without social information over the first four trials, to allow them to establish some expectations of the lakes. This was useful for the q-learning modeling (described below), which explored the effect of previous experience on advice taking.

...

Social information was always accurate (indicating the lake with the greater number of expected fish) to avoid introduction of irrelevant differences between social information conditions, but was not always in line with the experience-based expectations of the participants, and did not always lead to immediate high rewards.

...

In the fishing task (Figure 1), participants (N = 1492) chose to fish at one of two lakes and subsequently received feedback about the number of fishes they caught. Participants made this choice 15 times for each pair of lakes. The number of fishes caught was randomly drawn from a normal distribution, with one lake having higher average yield than the other ($M_{\text{Good}}=5.5$, $M_{\text{Bad}}=4$), and both lakes having the same variance ($\sigma=1.7$). As these distributions overlapped, it was possible to receive high rewards from the bad lake and vice versa in some trials. At the end of the task, participants were paid a bonus that was determined by the total number of fishes they caught (20 pence per 50 fishes caught) (Average response time and reward for each block are in S10). The experiment was programmed in JavaScript, and the code is available in the open materials.

Each participant completed three blocks: control, observation and advice (order counter-balanced), with each block differing in the social information provided to participants after the 4th trial. Social information always recommended the good (high expected reward) lake. However, this was not immediately apparent to the participants, as their initial experience in the first 4 trials could go against the social information (see analysis of initial experience in the Supplementary Materials Figure S1).

The social learning literature discusses phenomena very similar to what the authors call “advice” using the term “teaching”. There are multiple theoretical models (e.g. Castro & Toro, 2014; Fogarty et al., 2011) that explore the evolutionary conditions under which tutors should give a pupil adaptive advice even if it comes with some cost to themselves. The introduction would benefit from some discussion about how advice relates to teaching and which evolutionary dynamics might underpin both.

- Thanks for this comment. We agree with the reviewer that teaching is a relevant phenomenon to mention and it was previously mentioned in the discussion of the paper. We have retained this section. However, we feel it is beyond the scope of

the paper to outline conditions which might favour advice-giving and whether these would be similar to teaching given the additional characteristics which potentially distinguish the two (i.e. whether advisor and recipient are related, cost to recipient of learning independently, benefit of following advice, cost and benefit of transmitting advice etc). Since our paper is primarily about social learning rather than advice-giving, we feel this might be a bit of a diversion from the core message of the paper, though we do appreciate there is a broad relevance to these ideas.

Relatedly, in the introduction, the authors should better justify why they included a measure of paranoia and not of any other personality trait that might modulate advice taking. At the moment, it appears rather ad-hoc and should be better embedded in theory.

- Thank you for this important point. Including paranoia was a core part of our study since paranoia is, by definition, characterised by a tendency to believe that others intend the person harm. This perception of intentional malice is at the heart of the phenomenon we wish to study, which is the extent to which the perception of others' intentions affects social learning. However, we accept that given this is a key point, this needs to be made as clear as possible and we have now revised the Introduction to make this clearer (Lines 71-73).

An exaggerated tendency to believe that others have malign intentions - and associated suspicion and mistrust - is the basis of paranoia [31,32], which is not solely a clinical category but also varies across a full spectrum of severity in the general population [31,33,34]. Given the increased tendency to believe that others have harmful intentions, even when true intentions are ambiguous [35,36], we might expect higher levels of paranoia to be associated with lower levels of trust in others and, consequently, a reduced tendency to follow advice.

The authors used simple reinforcement learning models and estimated individual-level parameters that modulate the value and inverse temperature after receiving social information. They fitted a single parameter per participant for rounds 6-15 which might hide more intricate learning dynamics. The bimodal distribution of the beta boost, for instance, might not reflect the real distribution of inter-individual variation, but might be the result of learning trajectories that change differently over time. In addition, it might be constructive to compare how learning parameters differ before and after social information is provided. Researchers have implemented time-varying learning parameters in similar models that allow a closer look at the temporal dynamics of learning strategies (e.g. Deffner et al., 2020).

- As social information was introduced only once in our design, we did not include in our model a dynamic effect of social information, such as was used by Deffner

et al. 2020. Our model tracks the dynamics of non-social information, i.e. the experienced outcomes, and assumes two effects of social information. The first is a more transient effect which assumes that the advice gives a single value boost to the advised lake. This effect is expected to diminish as participants keep on learning and experiencing the lakes' outcomes. The second effect is the boost to precision (beta value) which is expected to last over the entire learning block. Different combinations of these two effects can lead to different learning trajectories, as demonstrated in our simulations. Importantly, the beta-boost reflects a dynamic change in learning parameter, as it adds to the initial beta value (Lines 278-280).

- We demonstrated that, following the model fitting procedure, beta boost followed a bi-modal distribution. To characterize how these parameters affected learning, we examined the learning trajectories of high beta-boost and low beta-boost individuals. In the high beta-boost group we observe a relatively consistent trajectory, as these participants tended to keep on choosing the same option. In the low beta-boost group we observe a gradual learning trajectory, in line with the simulations we carried. However, we agree that this trajectory may include more elaborate learning dynamics, and revised the manuscript to reflect this notion. However, the main focus of our analysis is in differentiating between the two groups, which display a clear difference in long term adherence to social information (Lines 331-334).

... Both changes to precision and value took place only once in our model, immediately after receiving the social information, in line with our experimental design.

...

This pattern was distinct from the more variable pattern associated with lower values, which may include more elaborate learning trajectories which are dependent on the specific outcomes experienced by participants in this group.

The authors should also provide more information on how the computational model was fitted. Based on the code, it seems like the authors used some custom made function instead of relying on established statistical inference framework. This is of course ok, but in this case, it is even more important to provide enough details on the exact procedure.

- We added a detailed description of the model fitting procedure in the supplementary materials and in the main text Lines 184-186 . We did not change the scripts for model fitting which were included in the open materials during the initial submission of the paper.

Minor points:

Line 141: "Independent variables" instead of "dependent variables"

- Many thanks for catching this. It has now been corrected.

Literature cited:

Castro, L., & Toro, M. A. (2014). Cumulative cultural evolution: the role of teaching. *Journal of Theoretical Biology*, 347, 74-83.

Deffner, D., Kleinow, V., & McElreath, R. (2020). Dynamic social learning in temporally and spatially variable environments. *Royal Society open science*, 7(12), 200734.

Fogarty, L., Strimling, P., & Laland, K. N. (2011). The evolution of teaching. *Evolution: International Journal of Organic Evolution*, 65(10), 2760-2770.

Toyokawa, W., Whalen, A., & Laland, K. N. (2019). Social learning strategies regulate the wisdom and madness of interactive crowds. *Nature Human Behaviour*, 3(2), 183-193.

Appendix B

Dear Prof. Brosnan

Thank you for the conditional acceptance of the paper. Please find below our responses to the reviewer's comments. We added clarification in the revised manuscript in line with the reviewer's comment regarding capping of long trials.

At the end of this document we attached the revised manuscript, with changes highlighted.

Our replies to the reviewers' comments are provided in red. Any copied sections from the manuscript are given in blue.

Best wishes,

Uri Hertz, Vaughan Bell and Nichola Raihani

Associate Editor:

Board Member: 1

Comments to Author:

Thank you very much for your careful and detailed revisions and responses to the reviewer feedback. Like the reviewers I agree that you have thoroughly addressed the reviewers' comments and questions. However, Reviewer 2 raises two points in reference to additional clarification regarding your methods and treatment of your data that I believe are worth addressing to aid understanding of your protocols and potential replications of your methods.

Reviewer(s)' Comments to Author:

Referee: 2

Comments to the Author(s)

Thanks for the very thorough and clear revisions that addressed all of the comments I had! I do not doubt the validity of the computational analyses in any way, but I would recommend the authors to use one of the standard probabilistic programming frameworks (such as stan or PyMC3) to fit such computational models in the future, instead of their custom, in-house code. This would make the models more transparent to others and would also increase the chances that other researchers will build on this great work. I am excited to see this published soon!

Many thanks for this important suggestion. We hope we have addressed the need for transparency by making all our modelling code freely available online. However, we also

agree that standardisation has important benefits, and we agree that this is an important direction for future studies.

Referee: 1

Comments to the Author(s)

Thank you for all your clarifications and responses. A few minor issues remain:

In response to my point about advice being assumed to be an amalgamation of experiences, compared to a single observation, you say : " In the instructions, participants were informed that they were either receiving advice from or secretly observing the choice of an expert player. We used this language specifically to address the important confound you raised above. "

But it is not clear to me how that language does address the important confound I raise. It could be the case that they either receive advice (made up of a number of experiences) from an expert player, or observe just one experience of an expert player. For this reason, and to get a sense of what the participants may have assumed/understood from the instructions, I think it's necessary to include a screenshot of the exact wording and instructions that the participants saw, either in the main manuscript, or in the supplementary material. This is particularly useful for researchers if they are wanting to replicate or base their own experiments on your work in the future.

We agree that making our experiment available for other researchers for future replications and extensions is important. We therefore included the exact wording used in the experiment in Figure 1 in the main text, as well as uploading the full Javascript code of the experiment (which includes the full instructions given to participants) in the online materials, allowing future replications of our experiment and results.

The final point: you mention after the participant payment information "No participant was excluded from analysis, but single trials that took more than 20 seconds to complete were removed." This is not mentioned anywhere in the preregistration and I wonder what it was about 20 seconds that determined those trials were removed. A simple justifying statement would suffice, and ideally these data exclusion criteria are included in future preregistrations. e.g. can you confirm that your results wouldn't change if the cut off was 25 seconds, or 15 seconds, or no cut-off at all?

We did not anticipate response time above 20 seconds, following our small pilot study, and therefore did not use this criterion in the pre-registration. However, we observed in the data that on some trials participants took long time to respond. We therefore decided to exclude long trials to remove trials which we thought did not reflect genuine task performance. We set the cut off at 20 seconds, which removed 0.3% of trials (199 trials out of 67140 trials, as there were 45 trials per participant). These 199 long trials were distributed across all experimental-conditions, and across all learning trials. The online dataset includes all the trials, for future replication efforts, and exclusion was done in the analysis scripts (also available online).

This exclusion had a very small effect on the observed results. For example, in the immediate-effect analysis (H1 and H1a), which is a key finding in our paper, only one trial was removed from analysis of the immediate effect of social information (i.e. only one person took longer than 20 seconds to respond on one of the blocks on the trial immediately following social information), and therefore uncapped analysis of the immediate response revealed the same effects. Below is the summary of the effects for H1 (Table S1 in the supplementary materials), with the capped analysis on the left and the uncapped analysis on the right.

	Capped at 20 sec (Table S1)			Uncapped analysis		
	Chisq	Df	Pr(>Chisq)	Chisq	Df	Pr(>Chisq)
(Intercept)	16.08	1	<0.0001	16.06	1	<0.0001
Condition	45.85	2	<0.0001	45.93	2	<0.0001
RGPTS_persecution	0.18	1	0.67	0.18	1	0.67
Intelligence	0.0007	1	0.97	0.0005	1	0.98
Condition:RGPTS_persecution	6.18	2	0.045	6.21	2	0.044
Condition:Intelligence	1.24	2	0.54	1.26	2	0.53

We now included this in the main manuscript (line 125):

No participant was excluded from analysis, but single trials that took more than 20 seconds to complete were removed, resulting in the removal of 0.3% of trials (199 out of 67140) across all experimental conditions and learning trials. This procedure was not pre-registered, as we did not anticipate long response times based on pilot study. We decided to exclude these long trials to remove trials which we thought did not reflect genuine task performance. We note that an analysis without trial exclusion did not lead to any meaningful changes in the results.